# Lamin A/C-dependent chromatin architecture safeguards naïve pluripotency to prevent aberrant cardiovascular cell fate and function

Yinuo Wang ⑮[1,2], Adel Elsherbiny[1], Linda Kessler[1], Julio Cordero[1,2], Haojie Shi[1,2], Heike Serke[1,2], Olga Lityagina[1], Felix A. Trogisch ⑮[2,3], Mona Malek Mohammadi ⑮[2,3,7], Ibrahim El-Battrawy[4,8], Johannes Backs ⑮[2,5], Thomas Wieland ⑮[2,6], Joerg Heineke ⑮[2,3] & Gergana Dobreva ⑮[1,2] ✉

Tight control of cell fate choices is crucial for normal development. Here we show that lamin A/C plays a key role in chromatin organization in embryonic stem cells (ESCs), which safeguards naïve pluripotency and ensures proper cell fate choices during cardiogenesis. We report changes in chromatin compaction and localization of cardiac genes in *Lmna−/−* ESCs resulting in precocious activation of a transcriptional program promoting cardiomyocyte versus endothelial cell fate. This is accompanied by premature cardiomyocyte differentiation, cell cycle withdrawal and abnormal contractility. Gata4 is activated by lamin A/C loss and *Gata4* silencing or haploinsufficiency rescues the aberrant cardiovascular cell fate choices induced by lamin A/C deficiency. We uncover divergent functions of lamin A/C in naïve pluripotent stem cells and cardiomyocytes, which have distinct contributions to the transcriptional alterations of patients with *LMNA*-associated cardiomyopathy. We conclude that disruption of lamin A/C-dependent chromatin architecture in ESCs is a primary event in *LMNA* loss-of-function cardiomyopathy.

Maintenance and plasticity of nuclear architecture and high-order chromatin organization are essential for the establishment of cell identity during embryonic development[1–3], and dysregulation can result in human disease[4]. An important determinant of 3D chromatin architecture is the nuclear lamina, a proteinaceous meshwork of intermediate filaments, which is located at the inner nuclear membrane. The nuclear lamina provides mechanical and structural support to the nucleus[5] and anchors heterochromatin regions termed lamina-associated domains (LADs) at the nuclear periphery, thereby shaping higher order chromatin structure[6]. Chromatin-nuclear lamina dynamics play a key role in cell fate decisions by "locking" or "unlocking" genes conferring cell identity at the nuclear periphery[3,7].

[1]Department of Cardiovascular Genomics and Epigenomics, European Center for Angioscience (ECAS), Medical Faculty Mannheim, Heidelberg University, Mannheim, Germany. [2]German Centre for Cardiovascular Research (DZHK), Mannheim, Germany. [3]Department of Cardiovascular Physiology, European Center for Angioscience (ECAS), Medical Faculty Mannheim, Heidelberg University, Mannheim, Germany. [4]First Department of Medicine, Faculty of Medicine, University Medical Centre Mannheim (UMM), Heidelberg University, Mannheim, Germany. [5]Institute of Experimental Cardiology, Medical Faculty Heidelberg, Heidelberg University, Heidelberg, Germany. [6]Experimental Pharmacology, European Center for Angioscience, Medical Faculty Mannheim, Heidelberg University, Mannheim, Germany. [7]Present address: Institute of Physiology I, Life & Brain Center, Medical Faculty University of Bonn, Bonn, Germany. [8]Present address: Bergmannsheil Bochum, Medical Clinic II, Department of Cardiology and Angiology, Ruhr University, 44789 Bochum, Germany. ✉e-mail: Gergana.Dobreva@medma.uni-heidelberg.de

Importantly, loss of chromatin tethering to the nuclear lamina following HDAC3 deletion causes release of cardiomyocyte (CM)-specific gene regions from the nuclear periphery, leading to precocious CM differentiation and heart disease pathogenesis[8].

The nuclear lamina consists of a proteinaceous meshwork of intermediate filaments, the A and B-type lamins. Lamins are multifunctional proteins, which play important roles in nuclear structure, chromatin organization, gene positioning, DNA replication and repair, as well as cell proliferation, differentiation and stress responses[9,10]. Lamins are also key components of the LINC complex, which physically couples the nucleoskeleton with the cytoskeleton[11-13]. Decoupling this connection via abnormal lamin A/C expression increases sensitivity to mechanical stress and impairs mechanosensing, which is particularly important for tissues that experience high levels of mechanical stress[11-13].

Consistent with its important biological functions, mutations in the gene for lamin A/C (*LMNA*) result in multisystem disease phenotypes collectively referred to as laminopathies[14]. Many of these diseases affect primarily the heart, including dilated cardiomyopathy (DCM) and arrhythmogenic cardiomyopathy (ACM), diseases with poor prognosis and a high rate of sudden cardiac death due to malignant arrhythmias[13-15]. Unfortunately, there is currently no specific treatment for *LMNA* cardiomyopathies, underscoring the importance of understanding the molecular mechanisms underlying the pathogenesis of this disease.

Modeling of *LMNA* cardiomyopathies in vitro using induced pluripotent stem cell (iPSCs)-derived CMs revealed increased chromatin accessibility at LADs upon *LMNA* haploinsufficiency, resulting in PDGF pathway activation contributing to an arrhythmic phenotype[16]. However, analysis of the 3D genome organization in CMs of another *LMNA* haploinsufficient model, which also showed contractile alterations[17], did not detect pronounced changes in chromatin compartmentalization into open and closed states that could explain the altered transcriptional activity. This leaves the question open to whether lamin A/C-mediated 3D chromatin architecture plays a role in *LMNA* cardiomyopathies. A more recent study revealed disruption of peripheral chromatin and de-repression of non-myocyte lineages in CMs carrying pathogenic *LMNA* T10I mutation[18].

These studies were conducted in CMs as it has been long thought, that A-type lamins play an important function after commitment of cells to a particular differentiation pathway, such as CMs. However, a developmental origin of cardiac laminopathies have been suggested using mouse embryonic stem cells (mESCs) and a mouse model harboring *Lmna* p.H222P mutation, causing Emery-Dreifuss muscular dystrophy and cardiomyopathy[19]. LmnaH222P mutation resulted in impaired epithelial-to-mesenchymal transition of epiblast cells, which in turn caused CM differentiation defects.

Here, we show that lamin A/C plays a central role in 3D chromatin architecture in naïve pluripotent stem cells, which ensures proper cardiovascular cell fate, differentiation and function and pinpoints key molecular determinants at the roots of cardiomyopathies due to *LMNA* loss-of-function (LOF).

## Results

### Lmna deficiency leads to aberrant cardiovascular cell fate choices

Chromatin tethering to the nuclear lamina have been proposed to control cardiac lineage restriction[8], however, to what extent nuclear lamins are involved in this process remains unknown. We therefore studied the role of A- and B-type lamins in cardiovascular differentiation using undirected mouse embryonic stem cell (mESC) differentiation to generate different cardiovascular cell types, as well as directed differentiation into CMs. Interestingly, *Lmna-/-* mESC differentiated in embryoid bodies (EBs) started beating already at day 7 (d7), and we observed a significantly higher number of beating EBs at later

days of differentiation (Fig. 1a, b, Supplementary Fig. 1a–c). Consistent with this, we detected a significant increase in the CM marker genes *Tnnt2*, *Mlc2v*, *Mlc2a* in *Lmna-/-* as well as *Lmna+/-* EBs, while the endothelial cell (EC) marker genes *Pecam1* and *Flk1* were downregulated (Fig. 1c, d). A similar increase in the expression of CM marker genes upon lamin A/C depletion was observed using directed differentiation of mESCs in CMs (Supplementary Fig. 1d).

To gain a better understanding of the mechanisms underlying lamin A/C function in cardiogenesis, we analyzed cardiac mesoderm commitment and cardiovascular progenitors diversification in more detail. Real-time PCR analysis for mesoderm (*Eomes*) and cardiac mesoderm markers (*Mesp1*) showed no significant difference in expression levels between control and lamin A/C-deficient EBs (Fig. 1e, Supplementary Fig. 1e). Similarly, FACS analysis for FLK-1 and PdgfR-α, as wells as for Nkx2−5 found no differences in early cardiovascular and cardiac progenitor (CP) numbers (Fig. 1f, g, Supplementary Fig. 1f). However, at the CP stage (d6) we detected significant upregulation of core cardiac transcription factors (TFs) (Fig. 1h), followed by increase in CM and decrease of EC numbers at later stages (Fig. 1i, Supplementary Fig. 1g), suggesting a role for lamin A/C in regulating cardiovascular progenitor cell fate.

To further study the mechanisms underlying cardiac cell fate choices upon lamin A/C loss we performed RNA-sequencing (RNA-Seq) of sorted cardiac progenitors (Fig. 1j, Supplementary Data 1a). Gene ontology (GO) analysis of genes upregulated in *Lmna-/-* CP cells revealed over-representation of GO terms linked to heart and cardiac muscle tissue development, including many key cardiac TFs, as well as genes involved in cardiac contraction and regulation of biosynthetic processes (Fig. 1j). In contrast, genes downregulated upon lamin A/C loss were enriched in GO terms linked to anterior/posterior patterning, alternative cell fates including vasculature development (Fig. 1j), consistent with the decreased number of ECs. Similarly, RNA-Seq analysis of d10 EBs revealed that genes involved in vascular development were decreased, whereas genes involved in heart contraction and calcium ion transport showed elevated expression (Supplementary Fig. 1h, i, Supplementary Data 1c).

To study whether the increased expression of cardiac structural and contraction genes is simply due to the increased number of CMs or if *Lmna−/−* CMs express these genes in higher level compared to control cells, we performed RNA-Sequencing of sorted Nkx2−5+ CMs. GO analysis revealed that upregulated genes in CMs were similarly enriched in GO terms linked to cardiac muscle contraction, regulation of heart rate as well as calcium ion transport, while downregulated genes were linked to cell cycle, anterior/posterior patterning, and alternative cell fates (Fig. 1k, Supplementary Data 1b).

In contrast to lamin A/C deficiency, ESC lines harboring the Lmna p.H222P mutation, causing Emery-Dreifuss muscular dystrophy and cardiomyopathy[20], and the Lmna p.G609G mutation, resulting in Hutchinson-Gilford progeria syndrome (accelerated aging and premature death due to cardiovascular events)[21,22] showed different differentiation behavior (Supplementary Fig. 2a–f). We observed significantly decreased expression of cardiac mesoderm marker genes, such as *Eomes* and *Mesp1*, at mesoderm stage (d4) (Supplementary Fig. 2c), cardiac progenitor (CP) markers at CP stage (Supplementary Fig. 2d) and CM genes at day 8 (Supplementary Fig. 2e). Consistent with the expression data we observed significantly decreased number of CMs (Supplementary Fig. 2f). These data are in line with previously published study, which showed that ESC lines harboring a Lmna p.H222P mutation have impaired cardiac differentiation[19], and suggest that the molecular mechanisms resulting in heart disease as a result of distinct point mutation in the *Lmna* gene are different. In contrast to lamin A/C deficiency or mutation, ablation of lamin B1 in mESCs did not largely affect CM differentiation (Supplementary Fig. 2g–k), suggesting that lamin A/C rather than lamin B1 plays a key role in cardiac lineage restriction.

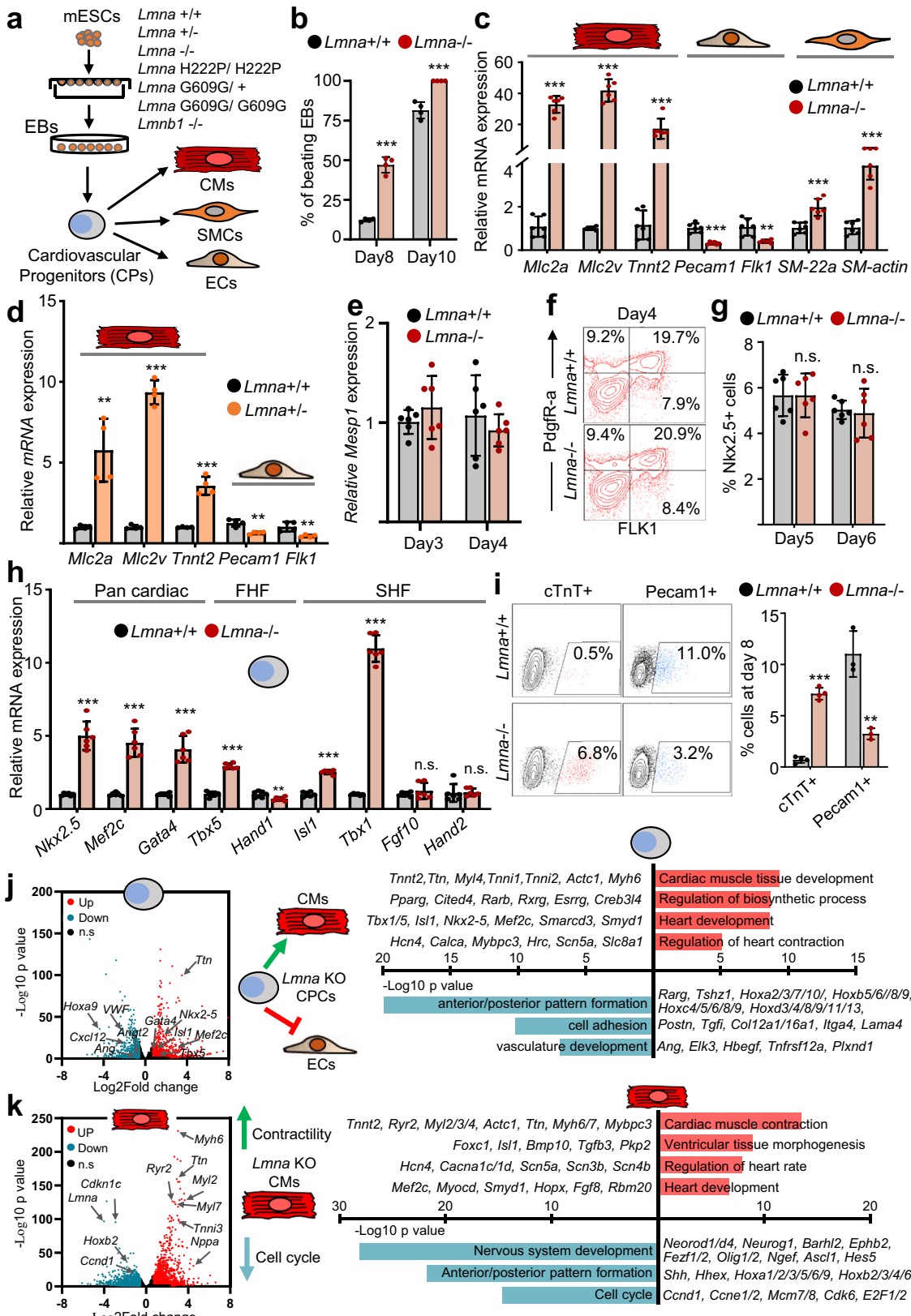

**Lamin A/C loss induces 3D chromatin reorganization in mESCs**

Since we detected major changes in gene expression in *Lmna-/-* versus *Lmna+/+* cardiac precursors, which have low lamin A/C levels, we analyzed lamin A/C levels in more detail. Lamin A/C was expressed in mESC cells[23], absent after exit of pluripotency and during early differentiation, and showed high levels in CMs (Fig. 2a). Consistent with

lamin A/C expression in mESCs, we observed dramatic changes in nuclear structure in mESCs lacking lamin A/C. *Lmna-/-* nuclei were bigger, showed irregular shape and enlarged nucleoli (Fig. 2b, Supplementary Fig. 3a). Lamin A/C-deficient mESCs formed flatter and more irregular colonies with varying levels of the pluripotency marker Oct4, in comparison to the compact and round control and lamin B1-

**Fig. 1 | Aberrant cardiovascular cell fate choices and precocious cardiomyocyte differentiation upon lamin A/C loss. a** Schematic diagram of the experimental setup. CMs, cardiomyocytes; SMCs, smooth muscle cells; ECs, endothelial cells. **b** Percentage of beating EBs derived from *Lmna*+/+ (*n* = 4) and *Lmna*−/− (*n* = 4) mESCs at day 8 (d8) and day 10 (d10). **c, d** qPCR analysis of CM (*Mlc2a*, *Mlc2v*, *Tnnt2*), endothelial (*CD31*, *Flk1*) and smooth muscle (*SM-22α*, *SM-actin*) genes in d10 *Lmna*+/+ and *Lmna*-/- EBs (**c**, *n* = 6) or *Lmna*+/+ and *Lmna*+/- EBs (**d**, *n* = 4). **e** Relative *Mesp1* expression in EBs differentiated from control and *Lmna*−/− mESCs at different days (*n* = 6). **f** Representative FACS analyses of PdgfR-α + /FLK1 + cardiovascular precursors in d4 control and *Lmna*−/− EBs. **g** Percentage of Nkx2-5+ cardiac precursors in d5 and d6 control and *Lmna*−/− EBs determined by FACS analysis (*n* = 6). **h** Relative mRNA expression of cardiac progenitor marker genes in d6 EBs (*n* = 6). FHF, first heart field; SHF, second heart field. **i** Representative FACS analyses of cardiac troponin (TnT)+ CMs and Pecam1+ ECs (left) and percentage of cTnT+ CMs (*n* = 4) and Pecam1+ ECs (*n* = 3) determined by FACS (right) at d8 of mESC differentiation. **j, k** Volcano plot showing the distribution of differentially expressed genes (DEG) between *Lmna*+/+ and *Lmna*−/− FACS-sorted Nkx2-5+ CPs from d6 EBs (**j**, left panel) and Nkx2-5+ CMs from d10 EBs (**k**, left panel). *n* = 3 biologically independent samples; log2 fold change ≤ −0.58, ≥0.58; *p*-value <0.05. *P*-value was calculated by DESeq2. Representative genes and enriched GO terms in upregulated (red) and downregulated (blue) genes are presented to the right. Significance is presented as *p*-values from pathway analysis using DAVID Bioinformatics Resources 6.8. **b**–**e**; **g**–**i**: Values for *n* represent biologically independent samples. Data are presented as mean ± SD and *p* values were determined by unpaired two-tailed Student's *t*-test. *P*-values are as follows: \*\*\**p* < 0.001, \*\**p* < 0.01, \**p* < 0.05. Source data are provided as a Source Data file.

deficient mESCs colonies which highly expressed Oct4 (Supplementary Fig. 3b, c).

Since we observed major nuclear enlargement in *Lmna-/-* mESCs, we next studied the effect of lamin A/C loss on the three-dimensional (3D) chromatin organization of mESCs using Hi-C. Interestingly, the Hi-C contact matrices of Chr. 14 were markedly different (Fig. 2c, Supplementary Fig. 3d, e), whereas we did not observe major changes in other chromosomes (Supplementary Fig. 3d). FISH analysis with Chr. 14 painting probes showed a significantly enlarged Chr. 14 territory, suggesting an important role of lamin A/C in Chr. 14 compaction (Fig. 2d, Supplementary Fig. 1c). More detailed analysis revealed a decrease in trans-interactions (involving different chromosomes) and an increase of long distance cis-interactions in *Lmna-/-* mESCs (Fig. 2e).

Previous studies demonstrated that the genome is organized into relatively active and inactive regions, referred to as A and B compartments, respectively[24]. Interestingly, around 8% of the chromatin compartments switched from A to B and vice versa as a result of lamin A/C depletion (Fig. 2f, Supplementary Data 2). GO analysis of genes located in inactive B compartments that transitioned to active A compartments in *Lmna-/-* mESCs were enriched for GO terms linked to calcium ion transmembrane transport, chromatin organization, muscle cell differentiation and relaxation of cardiac muscle, including genes such as *Myl4*, *Atp2a3*, *Ryr2*, *Camk2d*, etc. (Fig. 2g, h, Supplementary Fig. 3g). Active chromatin compartments that transitioned to an inactive state were enriched for genes linked to protein phosphorylation, nervous system development, cell migration and adhesion (Fig. 2g). Chr. 14 showed the most changes with around 13% active to inactive or vice versa transitions, consistent with the large scale chromatin decompaction at this chromosome (Fig. 2h, Supplementary Fig. 3f).

Using lamin A-DamID and lamin B1-DamID data from murine mESC[3,25], which map chromatin domains associated with lamin A and lamin B1, also referred to as lamin A LADs and lamin B1 LADs, we found a strong overlap of lamin A LADs as well as common Lamin A and B1 LADs with the compartments changes observed in *Lmna-/-* mESCs (Fig. 2i). Intersection of the genes found within lamin A-DamID and lamin B1-DamID data, revealed that 92% of the genes found in lamin B1 LADs were in lamin A LADs, whereas 6524 genes were uniquely found in lamin A LADs (Supplementary Fig. 3h). Within genes localized specifically in lamin A LADs were key cardiac TFs, such as *Gata4/6*, *Isl1* as well as cardiac structural proteins, such as *Actc1*, *Myl4*, *Mylk*, etc, whereas *Ryr2*, *Mef2c*, *Hcn1*, *Calca*, *Camk2d*, etc. were found in close proximity to both lamin A and B1 LADs (Fig. 3d).

Nuclear lamins tether chromatin at the nuclear periphery, which is characterized by low transcriptional activity. To analyze whether lamin A/C loss leads to dissociation of cardiac-specific genes from the repressive nuclear periphery already in mESCs we performed 3D FISH with probes for *Gata4*, *Mef2c*, *Actc1*, *Ttn* and *Kcnq1*. Interestingly, we observed a shift of CM-specific genes from the nuclear periphery to the active nuclear interior for genes found specifically in lamin A LADs as well as for genes found in lamin A/B1 LADs, whereas *Kcnq1*, which is not

found in LADs, was not affected (Fig. 2j–l), suggesting a role of lamin A/C in tethering cardiac genes to the repressive nuclear periphery in mESCs. In sharp contrast, CM-specific genes were still localized at the nuclear periphery in ESC lines carrying Lmna p.H222P and Lmna p.G609G mutations, suggesting that dissociation of CM genes from the nuclear periphery might be important for the precocious CM differentiation only observed upon lamin A/C ablation (Supplementary Fig. 3i, j).

## Lamin A/C deficiency in ESCs primes cardiac-specific gene expression

Further transcriptome analysis revealed a significant upregulation of genes associated with the nuclear lamina and encoding for major TFs and signaling molecules involved in stem cell differentiation, consistent with the primed state of *Lmna-/-* mESCs (Fig. 3a–c, Supplementary Data 3). Intriguingly, heart contractile proteins and Ca²⁺ transporters were also already transcriptionally upregulated in mESCs upon lamin A/C loss, consistent with the chromatin organization changes observed at these genes (Fig. 3a–c, Supplementary Data 3). In line with the transcriptional up-regulation of cardiac structural components such as *Myl4* and core cardiac TFs such as *Gata4*, we observed greatly increased MYL4 and GATA4 protein expression in lamin A/C-deficient mESCs (Fig. 3c).

Cluster analysis revealed that a large set of genes associated with lamin A LADs in mESCs were either upregulated in all stages of cardiac differentiation (Fig. 3d, e, cluster A) or specifically in mESCs, CPs or CMs (Fig. 3e, clusters B to G). These genes were linked to stem cell differentiation, heart development, calcium ion transport, as well as Wnt signaling (Fig. 3e; Supplementary Fig. 4a). Interestingly, we found a high overlap of genes associated to lamin A LADs in mESCs and genes that are upregulated in *Lmna-/-* CPs and CMs linked to heart morphogenesis and stem cell differentiation (Fig. 3e; Supplementary Fig. 4a), suggesting that lamin A/C deficiency in mESCs primes cardiac specific genes for expression in later stages during development.

To test this hypothesis, we next analyzed chromatin accessibility in wild-type and *Lmna-/-* ESCs using ATAC-Seq (Fig. 3f; Supplementary Fig. 4b–d). We observed a widespread increase in chromatin accessibility across the genome, as well as at genes upregulated upon lamin A/C LOF (Fig. 3f–h, Supplementary Fig. 4e, 4f, Supplementary Data 4). In contrast, there was little change in chromatin accessibility for genes that were downregulated in *Lmna-/-* ESCs (Fig. 3g, right panel). Consistent with the wide-spread chromosome 14 (Chr. 14) decompaction, we observed substantial accumulation of ATAC sequence peaks in Chr. 14 (Fig. 3f; Supplementary Fig. 4e), where *Gata4*, *Bmp4*, *Wnt5a*, *Myh6* and *Myh7* are located. The increased chromatin accessibility was accompanied by an upregulation of a large number of genes within this chromosome region not only in *Lmna-/-* ESCs, but also in *Lmna-/-* CPs and *Lmna-/-* CMs (Fig. 3f).

Cluster analysis revealed that a large set of genes that showed increased chromatin accessibility in ESCs were either upregulated in all stages of cardiac differentiation (Fig. 3i, cluster A) or specifically in

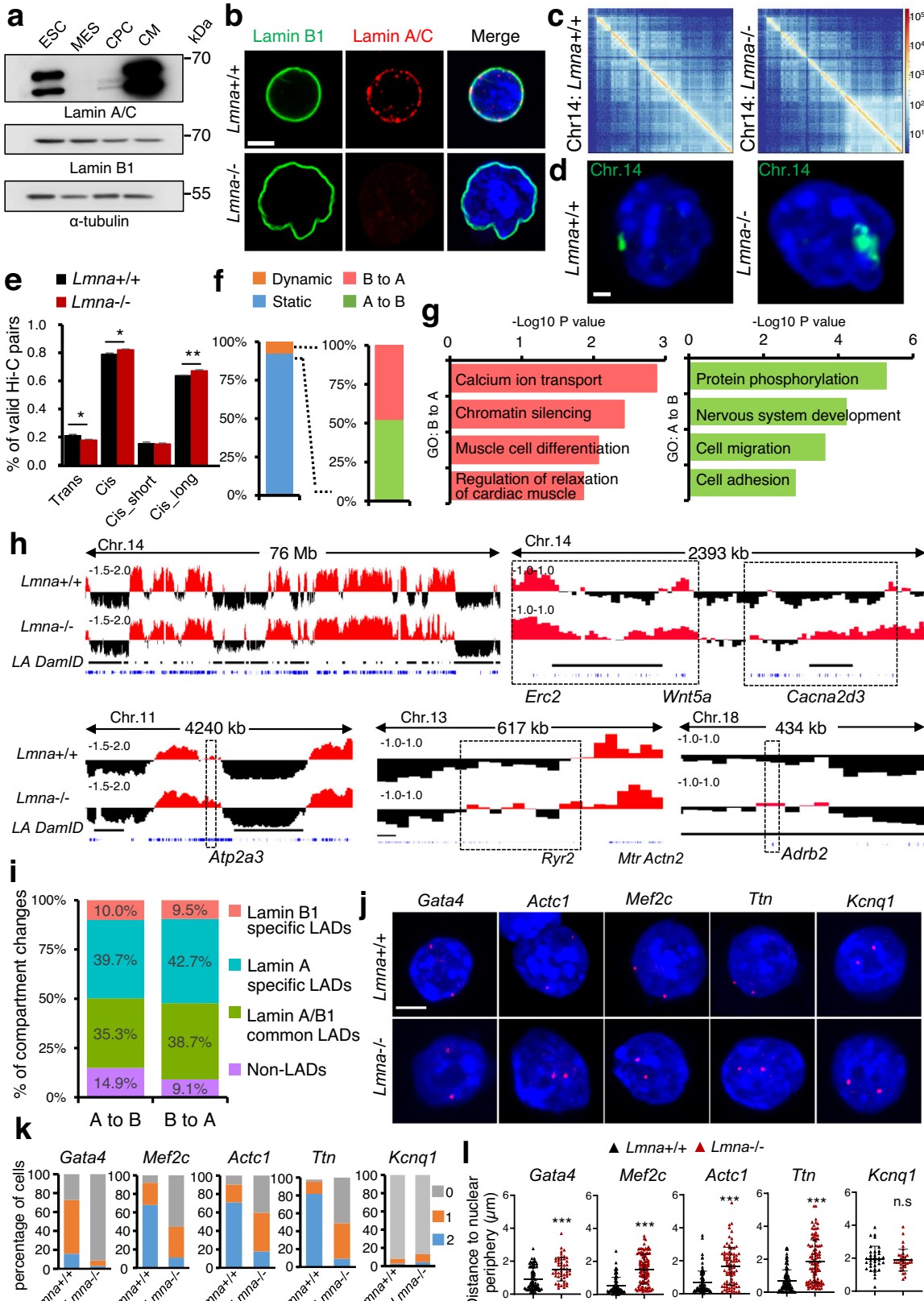

ESCs, CPs or CMs (Fig. 3i, clusters B-G). Intersection of the ATAC-Seq and RNA-Seq analysis revealed that more than half of the genes upregulated upon lamin A/C loss in ESCs showed increased chromatin accessibility (Fig. 3j) and GO analysis of the overlapping genes indicated overrepresentation of GO terms linked to transcription, cell differentiation, outflow tract morphogenesis as well as Wnt signaling

(Fig. 3k). Motif enrichment analysis within chromatin regions of genes upregulated upon lamin A/C LOF and characterized with more open chromatin identified binding motifs for Klf, Sox and Tead family members as well as Gata4 binding motifs (Fig. 3l).

Interestingly, intersection of ATAC-Seq in ESCs with the RNA-Seq analysis from CPs and CMs, revealed a high overlap of genes that

**Fig. 2 | Alterations in 3D chromatin organization in Lmna-/- mESCs. a** Western blot analysis of lamin A/C and lamin B1 during directed CM differentiation. **b** Confocal images of immunostainings for lamin A/C (red), lamin B1 (green), and DAPI (nucleus, blue) of *Lmna*+/+ and *Lmna*−/− mESCs. Scale bar, 10 μm. **c** Representative log transformed contact matrices of Chr. 14 at 100 kb resolution in *Lmna*+/+ and *Lmna*−/− mESCs. **d** FISH chromosome painting of Chr. 14 (green) in control and *Lmna*−/− mESCs. **e** Percentage of trans or cis interactions and cis interactions with distances less or more than 20 kb (cis_short or cis_long, respectively). **f** Genomic regions divided into static, i.e., without compartment transitions and dynamic, with A/B compartment transitions (left graph). Dynamic regions are subdivided into A to B and B to A transitions (right graph). **g** GO analysis of genes showing B to A and A to B transitions. **h** Genome tracks of Hi-C PC1 at 25 kb resolution and lamin A DamID regions at genes upregulated in *Lmna*-/- mESCs showing A (red) and B (black) compartments in *Lmna*+/+ and *Lmna*−/− mESC.

Lamin A/C DamID regions are presented as black boxes below the tracks. **i** Percentage of compartment changes showing overlap with lamin A-specific LADs, lamin B1-specific LADs and lamin A/lamin B1-common LADs as well as non-LADs. **j** Representative DNA FISH images of genes (red) within LADs and DAPI (blue) showing re-localization from the nuclear periphery to the nuclear interior in *Lmna* +/+ or *Lmna*−/− mESCs. *Kcnq1*, not found in LADs, did not show relocalization. Scale bar, 4 μm. **k** Quantification of the percentage of cells showing zero, one, or two alleles at the nuclear periphery for the indicated genes. **l** Quantification of the distance of *Gata4*, *Mef2c*, *Actc1*, *Ttn*, and *Kcnq1* to nuclear periphery in individual nuclei of *Lmna*+/+ and *Lmna*−/− mESCs. **c**–**i**: $n = 3$ biologically independent samples. **k**, **l**: $n = 30$-50 cells were quantified from three independent experiments. Data are presented as mean ± SD and p values were determined by unpaired two-tailed Student's *t*-test. *P*-values are as follows: \*\*\*$p < 0.001$, \*\*$p < 0.01$, \*$p < 0.05$. Source data are provided as a Source Data file.

undergo chromatin opening as a result of lamin A/C loss in ESCs and genes that are upregulated only later during differentiation, i.e., in *Lmna*-/- CPs and CMs (Fig. 3m). GO analysis of genes showing open chromatin conformation already at the ESC stage and upregulation in CPs and CMs revealed overrepresentation of GO terms linked to heart morphogenesis and cell fate commitment (Fig. 3n), supporting the notion that lamin A/C deficiency in ESCs primes cardiac specific genes for expression in later stages during development. Motif enrichment analysis within chromatin regions of genes upregulated upon lamin A/C LOF in CPs and CMs showing increase in chromatin accessibility (epigenetic priming) already in ESCs, identified binding motifs for Tead and Gata family members as well as Meis1, which regulates postnatal CM cell cycle arrest[26] and Foxo1, shown to be activated and to contribute to the pathogenesis of cardiac laminopathies[27] (Fig. 3o). Genes, which were downregulated upon lamin A/C deficiency and showed decreased chromatin accessibility were linked to cell adhesion, cell migration and angiogenesis (Supplementary Fig. 4g, 4h).

In contrast, *Lmna*^H222P/H222P, *Lmna*^G609G/+ and *Lmna*^G609G/G609G ESCs did not show obvious increase in genome-wide chromatin accessibility or at Chr. 14 as well as altered expression of genes that were deregulated upon lamin A/C LOF, supporting further the notion that the molecular mechanisms underlying the cellular phenotypes upon lamin A/C LOF and mutation are different (Supplementary Fig. 4i–k). Further, depletion of lamin A but not lamin C resulted in loss of tethering of CM-specific genes at the nuclear periphery and upregulation of lamin A/C-target genes (Supplementary Fig. 4l–o), suggesting the lamin A rather than lamin C plays an important role in cardiac lineage restriction.

## Cell-type specific role of Lamin A/C in shaping chromatin accessibility

We next studied the specific role of lamin A/C on chromatin architecture in CMs (Fig. 4, Supplementary Fig. 5a–d). Consistent with our ESCs results, we observed an increase in chromatin accessibility at Chr. 14 as well as genes upregulated in *Lmna*-/- CMs (Fig. 4a, Supplementary Fig. 5d, Supplementary Data 5), which belonged to GO terms related to heart morphogenesis, regulation of heart rate and contraction as well as calcium ion homeostasis, such as *Nkx2–5*, *Bmp10*, *Adrb1*, *Actc1*, *Ttn*, *Mybpc3*, *Myl7*, etc. (Fig. 4c, d). In contrast, genes which were downregulated upon lamin A/C-deficiency and showed deceased chromatin accessibility, were linked to cell differentiation and anterior/posterior patterning (Supplementary Fig. 5e, 5f). Motif enrichment analysis within chromatin regions of genes upregulated upon lamin A/C LOF in CMs identified binding motifs for Mef2c, Isl1, Gata, Tead, Tbx family members and many other key cardiac TFs promoting CM differentiation as well as Meis1 and Foxo1 (Fig. 4e), consistent with the functional changes observed in lamin A/C-deficient CMs.

We next compared the role of lamin A/C for chromatin organization in CMs to that in ESCs. Importantly, we found 55% overlap between genes that showed increased chromatin accessibility in both *Lmna*-/- CMs and ESCs, whereas 45% showed more open chromatin

only in CMs (Fig. 4f, Supplementary Fig. 5g). Genes that showed increased chromatin accessibility only in *Lmna*-/- ESCs but were upregulated in CMs and belonged to GO terms linked to cell fate commitment, whereas 530 genes that showed chromatin opening in both *Lmna*-/- ESCs and CMs were enriched in GO terms linked to cardiac muscle contraction and cellular calcium ion homeostasis. The 414 genes which showed specific chromatin opening upon lamin A/C loss only in CMs were enriched for GO terms associated with cardiac muscle contraction and sarcomere organization (Fig. 4f, Supplementary Fig. 5g).

Further analysis revealed that genes within lamin A LADs that showed chromatin opening upon lamin A/C loss specifically in ESCs were linked to cell aging and metabolism, whereas genes showing increased chromatin accessibility in both ESCs and CMs were linked to heart development and alternative cell fates (Fig. 4g). Interestingly, among the genes showing specific chromatin opening in CMs were *Ttn* and *Actc1*, both found in lamin A/lamin B1 LADs which repositioned from the nuclear periphery to the nuclear interior in ESCs, suggesting that only gene repositioning from the nuclear lamina toward the interior is not sufficient to induce chromatin opening. In addition, 32% of the genes upregulated in CMs were within LADs, with 23% specifically within lamin A LADs and 8% within lamin A/B1 LADs, supporting a dominant function of lamin A/C in cardiogenesis (Fig. 4h).

Further, we found that 26% of downregulated genes in *Lmna*-/- CMs were in lamin A LADs whereas 14% in lamin A/B1 LADs in ESCs, supporting a direct role of lamin A/C in chromatin reorganization and transcriptional activation particularly at genes related to cell cycle and anterior/posterior patterning (Supplementary Fig. 5h). Taken together our data indicate that lamin A/C plays a key role in transcriptional priming of genes involved in cardiac development, CM contraction and Ca2+ homeostasis in ESCs, whereas in CMs lamin A/C specifically regulates genes involved in cardiac contraction and sarcomere organization.

## Lamin A/C is crucial for naïve pluripotency

Since our results revealed an important function of Lamin A/C in mESCs for cardiogenesis, we next studied Lamin A/C expression in more detail. During mouse embryogenesis, *Lmna* mRNA levels were low in zygote and during cleavage, but were higher in the inner cell mass and mESCs and increased in CMs (Fig. 5a). Similarly, during human embryogenesis *LMNA* levels were higher in the inner cell mass and human-induced pluripotent stem cells (hiPSCs)-derived CMs (hiPSCs-CMs), however they were low in hiPSCs (Fig. 5a, b).

Human iPSCs cultured in standard conditions represent a primed state[18], thus we next examined lamin A/C levels and function in naïve versus primed pluripotency. Interestingly, primed mESCs did not express detectable levels of lamin A/C protein compared with naïve mESCs (Fig. 5c) similarly to human iPSCs cultured in standard conditions, while human iPSCs cultured in naïve state conditions highly

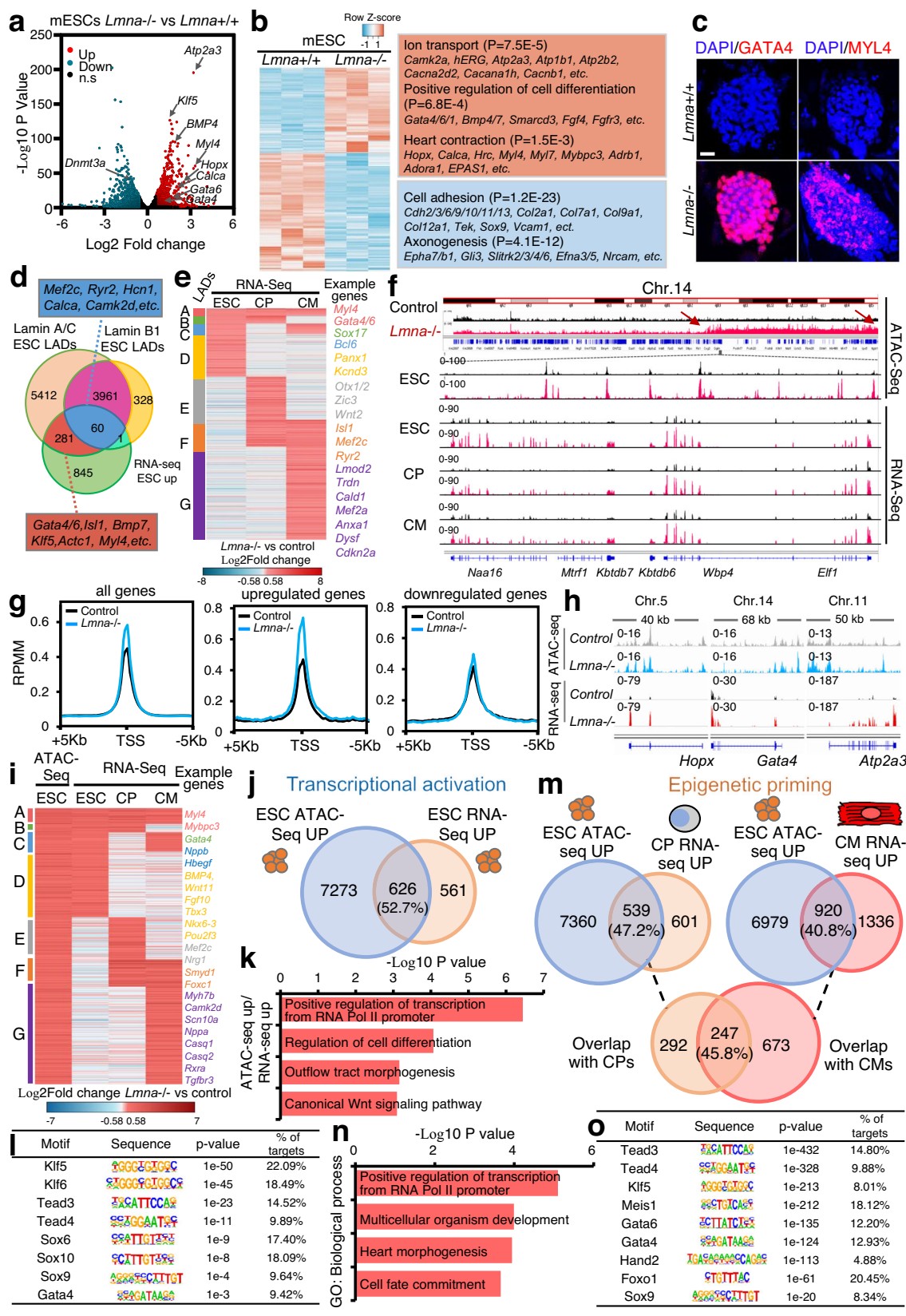

expressed lamin A/C (Fig. 5d, Supplementary Fig. 6a, b). Lamin A/C-deficient human iPSCs failed to establish naïve pluripotency due to poor cell survival while silencing of lamin A/C in naïve iPSCs resulted in flatter colonies, suggesting an important role of Lamin A/C in establishment and maintenance of naïve pluripotency (Fig. 5e, Supplementary Fig. 6c, d).

## Distinct roles of lamin A/C in naïve PSC and CMs for cardiac laminopathies

We next investigated whether the functional changes in cardiovascular cells following loss of lamin A/C were due to its role in the suppression of cardiac gene expression program in naïve mESCs. Similarly to ablation of *Lmna*, knockdown of lamin A/C in mESCs using shRNA led

**Fig. 3 | Transcriptional activation and epigenetic priming in mESCs upon lamin A/C loss results in cardiac programming. a** Volcano plot showing the distribution of DEG between *Lmna*+/+ and *Lmna*−/− mESCs (log2 fold change ≤−0.58, ≥0.58; *p*-value <0.05, determined by DESeq2 algorithm). **b** Heat-map representation of RNA-Seq analysis of control and *Lmna*-/- mESCs. Representative genes and enriched GO terms in upregulated (red) and downregulated (blue) genes are presented. **c** Confocal images of immunostaining for GATA4, MYL4 (red), and DAPI (blue) of *Lmna*+/+ and *Lmna*−/− mESCs. Scale bars, 20 μm. **d** Overlap of genes within lamin A LADs, lamin B1 LADs and genes upregulated in *Lmna*−/− mESCs. **e** Hierarchical cluster analysis of genes within lamin A LADs and upregulated either in *Lmna*−/− mESCs, *Lmna*−/− CPs, or *Lmna*−/− CMs. **f** Genome tracks of ATAC-Seq reads in *Lmna*+/+ and *Lmna*−/− ESCs and normalized RNA-Seq reads in control (black) and *Lmna*−/− (red) ESCs, CPs and CMs at Chr. 14. Red arrows indicate large-scale chromatin opening. **g** Normalized ATAC-Seq signal intensity at the TSS ± 5 kb of all genes (left) as well as genes upregulated (middle) and downregulated (right) upon lamin A/C LOF in *Lmna*+/+ and *Lmna*−/− ESCs. **h** Genome tracks of merged ATAC-Seq and RNA-Seq reads of *Lmna*+/+ and *Lmna*−/− ESCs at example genes. **i** Hierarchical cluster analysis of genes showing increased chromatin accessibility in ESCs and upregulation either in *Lmna*−/− ESCs, *Lmna*−/− CPs or *Lmna*−/− CMs. **j, m** Overlap between genes showing increased ATAC-Seq signal and genes upregulated upon *Lmna* LOF in ESCs (**j**) or in CPs and CMs (**m**). **k, n** GO analysis of the overlapping genes in (**j** and **m**, respectively). **l, o** Enrichment of known TF motifs in ATAC-Seq peaks of genes showing increased chromatin accessibility and upregulation in *Lmna*−/− ESCs (**l**) or upregulation in both *Lmna*−/− CPs and CMs (**o**). **a-o:** ATAC-Seq and RNA-Seq datasets from control and *Lmna*−/− ESCs, CPs, and CMs are from *n* = 3 biologically independent samples. **b, k, n** *P*-values are from GO pathway analysis using DAVID Bioinformatics Resources 6.8. **l, o** *P*-values were determined using Homer motif enrichment analysis.

to precocious CM differentiation and significantly higher expression of key cardiac TFs, as well as CM marker genes (Fig. 5f top left panel, Supplementary Fig. 6e, f). In contrast, depletion of lamin A/C in CPs or CMs led to a significant decrease of CM-specific gene expression (Fig. 5f top right and bottom left panels, Supplementary Fig. 6g, h). RNA-sequencing of CMs with CM-specific depletion of lamin A/C revealed upregulation of multilineage-specific TFs, mitogen-activated protein kinase (MAPK) as well as apoptosis-related molecules. Consistent with our qPCR analysis, many cardiac muscle contraction genes were downregulated, as well as genes involved in angiogenesis and cell migration (Fig. 5g). Similarly, lamin A/C knockdown in mouse neonatal CMs led to decrease in the expression of CM marker genes (Fig. 5f right bottom panel, Supplementary Fig. 6i). Interestingly, 68% of lamin A/C depleted mouse neonatal CMs showed abnormal proarrhythmic $Ca^{2+}$ transients, consistent with the role of lamin A/C in regulating chromatin conformation of genes involved in $Ca^{2+}$ homeostasis in both mESCs and CMs (Supplementary Fig. 6j–l).

Since we observed differences upon lamin A/C depletion in mESCs or specifically in CMs, we next compared the transcriptome of lamin A/C depletion specifically in CMs with the expression data from patients with *LMNA*-associated DCM, characterized by significantly lower lamin A/C protein levels[28]. Interestingly, GO analysis revealed a different subset of overlapping genes (Fig. 5h, i), implying important distinct functions of lamin A/C in both mESCs and CMs for proper cardiogenesis and cardiac function. Genes upregulated in both *Lmna*-/- mESC-derived CMs and DCM patients were linked to muscle contraction (Fig. 5h), while genes upregulated in *Lmna* knockdown CMs and DCM patients were linked to intracellular signal transduction, JNK activity and ER stress (Fig. 5i). Interestingly, among the genes overlapping between patients with pathogenic *LMNA* mutations and upon lamin A/C depletion in mESCs but not in CMs we found many genes associated with DCM, e.g., *RBM20*, *LDB3*, *ACTN2*, *MYOM1*, *MYL2*, etc. Downregulated genes in both cases were enriched for genes linked to mitotic cell cycle and cell division (Fig. 5h, i).

Consistent with these data, silencing of lamin A/C in naïve hiPSCs resulted in upregulation of cardiac structural and contraction genes (Fig. 5j), while silencing of lamin A/C in primed hiPSCs resulted in their downregulation (Fig. 5k). In line with the transcriptomics analysis, genes involved in cell cycle were similarly downregulated in both cases (Supplementary Fig. 6m–o). Taken together, these results on the one hand suggest that lamin A/C function in naïve pluripotent stem cells contributes to the pathological phenotype resulting from lamin A/C depletion, and on the other hand imply important distinct functions of lamin A/C in both mESCs and CMs for proper cardiogenesis and cardiac function.

### *Lmna* deficiency results in non-compaction cardiomyopathy

Next, we sought to corroborate our findings in vivo using *Lmna*+/+ (control), *Lmna*+/- and *Lmna*-/- mice[29]. Expression analysis revealed significant upregulation of CP and CM marker genes in dissected pharyngeal mesoderm and hearts of E8.5 embryos as well as E9.5 hearts upon *Lmna* ablation consistent with our in vitro cell culture based studies (Fig. 6a, b). Further, FISH analysis showed that CMs specific genes were located in the nuclear interior of *Lmna*+/+ CMs and at the nuclear periphery in *Lmna*+/+ fibroblasts respectively, while they were found in the nuclear interior in both CMs and fibroblasts isolated from E14.5 *Lmna*+/- and *Lmna*-/- embryos, supporting a role of lamin A/C in tethering CM specific genes to the nuclear periphery in non-CMs (Fig. 6c, d).

Functional analysis revealed significantly increased right ventricular ejection fraction and fractional shortening (RVEF and RVFS), an index of cardiac contractility, as well as lower RV systolic diameter in *Lmna*+/− and *Lmna*−/− embryos while the RV diastolic diameter was not different from the controls (Fig. 6e, f, Supplementary Fig. 7a, 7b). In contrast, we did not observe differences in LV function (Supplementary Fig. 7a, 7c). Histological examination unveiled non-compaction of the RV myocardium in both *Lmna*+/− and *Lmna*−/− embryos, while only *Lmna*−/− embryos showed LV non-compaction (Figs. 6g, 6h). Further, we observed significant decrease in CM proliferation in E18.5 *Lmna*+/- and *Lmna*-/- hearts using immunostaining for the mitotic marker phosphorylated-histone H3 (pH3) in combination with Troponin I (Figs. 6i, j). We also detected increased number of binucleated CMs via FACS analysis using Vybrant DyeCycle DNA dye[30] and immunostaining for DAPI and Troponin I (Fig. 6k–n), suggesting that precocious CM differentiation leads to premature binucleation and CM cell cycle withdrawal during fetal development. Moreover, consistent with the decreased number of ECs upon lamin A/C depletion observed in our in vitro system, we found significantly decreased capillary density in hearts from both *Lmna*+/- and *Lmna*−/− E18.5 embryos (Fig. 6o, p).

At birth *Lmna*+/− and *Lmna*−/− mice were indistinguishable from their control littler mates, but a week later there was a clear difference in size and body weight of *Lmna*-/- mice (Supplementary Fig. 7d). *Lmna*+/− and *Lmna*−/− neonatal mice showed increased RVEF, while LVEF was increased only in *Lmna*−/− mice (Fig. 7a, c), consistent with the more pronounced defects observed in the RV during embryogenesis. At P1 we observed significant right ventricular dilatation in *Lmna*+/− and *Lmna*−/− mice with no signs of pulmonary arterial hypertension (PAH) or pulmonary congestion (Supplementary Fig. 7e). The left ventricular posterior wall thickness at end-diastole (LVPW:d) was also significantly increased in both *Lmna*+/− and *Lmna*−/− mice, whereas left ventricle (LV) ejection fraction (EF) and LV fractional shortening (FS) was significantly increased only in *Lmna*−/− mice (Fig. 7c).

In sharp contrast, two weeks after birth the HW/BW was strongly decreased in *Lmna*−/− mice, and LVEF, LVFS and LVPW:d were reduced, while left ventricular end-diastolic area (LVEDA) was increased (Fig. 7d-f). No PAH or pulmonary congestion were observed at P14 (Supplementary Fig. 7f). Similar to the embryonic hearts, we

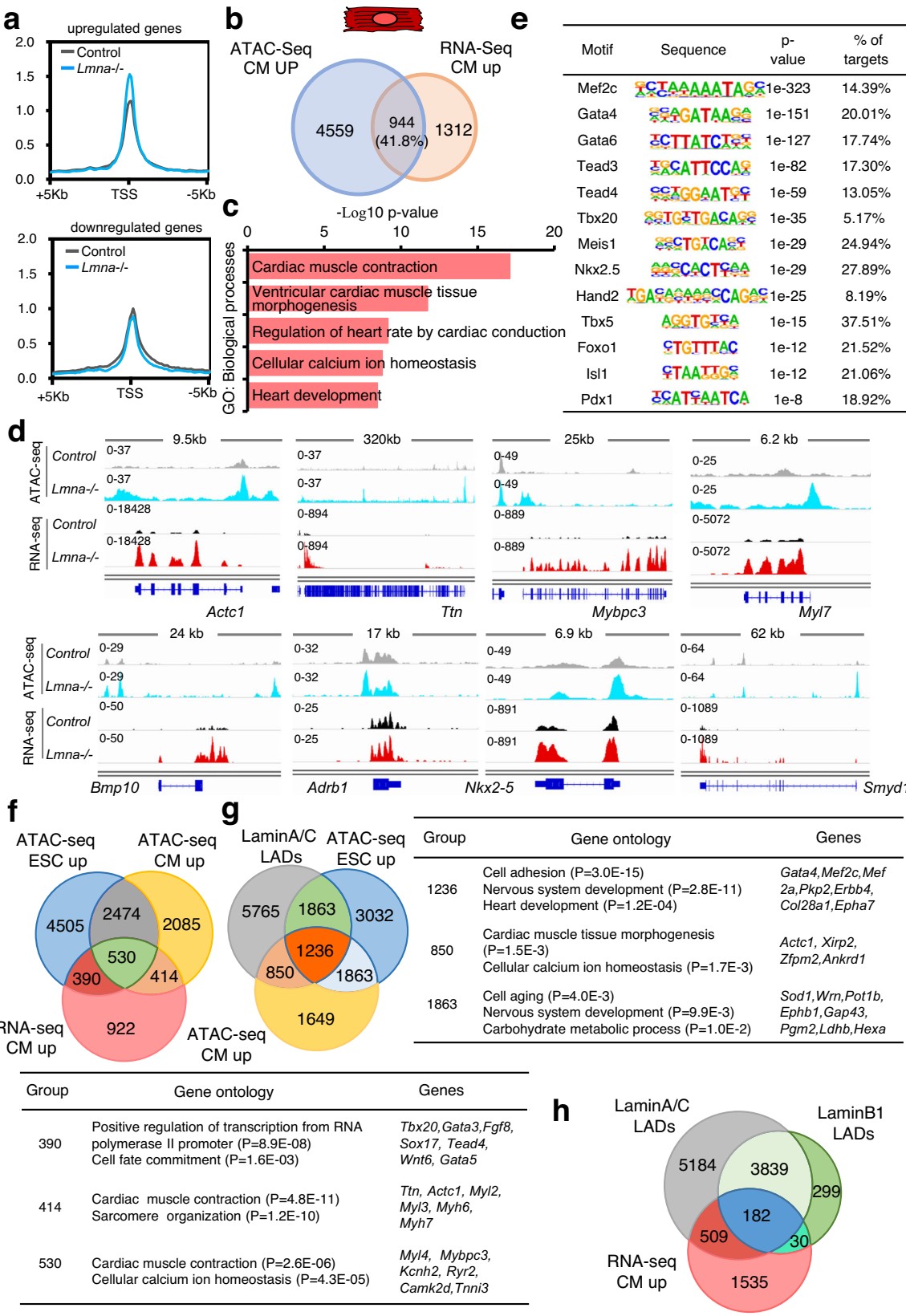

observed a decrease in capillary density in hearts from both *Lmna*+/- and *Lmna*-/- mice also at P1 (Fig. 7g, h). Furthermore, lamin A/C-deficient CMs showed reduced size in cross sections (Fig. 7i). Similar differences were also observed at P4 and P14 (Fig. 7j–l, Supplementary Fig. 7g–i). Immunostaining for the mitotic marker phosphorylated-

histone H3 (pH3) as well as the cytokinesis marker Aurora B in combination with Troponin I revealed significant decrease in CM proliferation at both P1 and P4 similar to the decreased CM proliferation observed in E18.5 embryonic hearts (Fig. 7m, n, Supplementary Fig. 7j–n).

**Fig. 4 | ESC- and CM-specific alterations of chromatin accessibility upon lamin A/C deficiency. a** Normalized ATAC-Seq signal intensity at TSS ± 5 kb of upregulated (top) and downregulated (bottom) genes in *Lmna*+/+ (gray) and *Lmna*−/− (blue) CMs. **b** Overlap of genes showing increased ATAC-Seq signal and upregulation in *Lmna*−/− CMs (n = 3; log2 fold change ≤−0.58, ≥0.58; *p*-value <0.05, determined by DESeq2 algorithm). **c** Representative GO terms within genes showing increased chromatin accessibility and expression in *Lmna*−/− CMs. **d** Examples of genes showing increased chromatin accessibility and upregulation in *Lmna*−/− CMs. Genome tracks of merged ATAC-Seq and RNA-Seq reads of *Lmna*+/+ and *Lmna*−/− CMs are presented. **e** Enrichment of known TF motifs in ATAC-Seq peaks of genes showing increased chromatin accessibility and upregulation in *Lmna*−/− CMs. **f** Overlap between genes upregulated in *Lmna*-/- CMs and genes showing increased ATAC-Seq signal in *Lmna*−/− ESCs and CMs (top). Enriched GO terms for the genes within the different overlaps are presented in a table (bottom). **g** Overlap between genes within lamin A LADs and genes showing increased ATAC-Seq signal in *Lmna*−/− ESCs and CMs (left). Enriched GO terms for the genes within the different overlaps are presented in a table (right). **h** Overlap between genes upregulated in *Lmna*−/− CMs and genes within lamin A and lamin B1 LADs in ESCs. **a**–**h**: ATAC-Seq and RNA-Seq datasets from control and *Lmna*-/- CMs, are from n = 3 biologically independent samples. **c**, **f**, **g** *p*-values are from GO pathway analysis using DAVID Bioinformatics Resources 6.8. **e** *P*-values were determined by Homer motif enrichment analysis.

## Premature binucleation, altered contractility, and increased cell death of *Lmna*−/− CMs

To study the effect of lamin A/C deficiency on CM function in more detail we utilized CMs isolated from P1 wild-type, *Lmna*+/- and *Lmna*−/− mice as well as CM differentiated from control and lamin A/C-deficient mESCs. A high percentage of *Lmna*+/- and *Lmna*−/− CMs were already binucleated at P1 similar to E18.5 hearts and we observed a significantly higher percentage of cells positive for γ-H2AX, a biomarker of DNA damage (Fig. 8a–d), suggesting precocious CM binucleation and increased DNA damage-induced cell death in *Lmna*+/− and *Lmna*−/− CMs.

Since we observed many $Ca^{2+}$ channels to be deregulated in *Lmna*-/- CM (Fig. 1j), we analyzed the intracellular $Ca^{2+}$-handling properties of wild-type, *Lmna*+/- and *Lmna*−/− CMs isolated from P1 mice. Although wild-type and most of *Lmna*+/- CMs showed uniform $Ca^{2+}$ transients, the peak height was significantly increased in *Lmna*+/- CMs and the time to peak was shorter (Fig. 8e, f), suggesting more rapid release of $Ca^{2+}$ from sarcoplasmic reticulum stores, consistent with the increased levels of Ryr2 (Fig. 1j, Supplementary Fig. 8c). In contrast, 65% of *Lmna*−/− CMs showed highly abnormal proarrhythmic $Ca^{2+}$ transients (Fig. 8e–g).

Similar functional alterations were observed in *Lmna*−/− mESC-derived CMs, which showed arrhythmic contractions and higher percentage of binucleated and γ-H2AX CMs (Fig. 8h–m, Supplementary Fig. 8a, b and 8d, e). Initially, the contraction amplitude of *Lmna*−/− mESC-derived CMs was much higher than control CMs but decreased with time, accompanied with extensive cell death (Supplementary Movie 1–4). Annexin V/7-AAD staining revealed a dramatic increase of apoptotic and dead *Lmna*−/− CMs in comparison to control CMs (Fig. 8n). The increased cell death was specific for lamin A/C-deficient CMs, because the non-CM population lacking lamin A/C as well as lamin A/C-deficient mESCs did not show increased cell death (Fig. 8n, Supplementary Fig. 8f).

Intriguingly, mechanical stretch or treatment of *Lmna*+/+ and *Lmna*−/− non-CMs with $H_2O_2$ resulted in dramatic increase of γ-H2AX and defects in DNA damage repair in *Lmna*−/− compared to control cells, suggesting that the premature cell cycle withdrawal and increased cell death in *Lmna*-/- CMs might be due to the inability of lamin A/C-deficient cells to respond adequately to mechanical and oxidative stress during early postnatal life (Fig. 8o, p)[31].

## Gata4 activation upon *Lmna* loss leads to aberrant cardiac development

Our results showed a significant upregulation of key cardiac TFs such as Gata4 and significant enrichment of Gata4 motifs at genes showing increase in chromatin accessibility upon lamin A/C loss (Fig. 3, Fig. 4). To investigate whether Gata4 upregulation might be responsible for aberrant cardiovascular cell fate choices and precocious CM differentiation upon lamin A/C loss, we decreased Gata4 levels using several different approaches: (1) siRNA mediated Gata4 silencing to transiently downregulate Gata4 in mESCs shortly before differentiation (Fig. 9a, b); (2) shRNA mediated knockdown carefully titrated to reduce Gata4 expression back to control levels in CPs (Fig. 9c); (3) ablation of one

Gata4 allele to reduce its expression (Fig. 9d–g). In all cases, we observed pronounced rescue of the *Lmna*−/− phenotype, i.e., a significant decrease of key cardiac TFs and reduced expression of CM marker genes (Fig. 9a, c, e). Furthermore, we detected a significant decrease of CM numbers and an increase of ECs in differentiating Gata4-depleted *Lmna*−/− mESCs compared to *Lmna*-/- mESCs (Fig. 9b, f, g, Supplementary Fig. 9b), demonstrating that decreasing Gata4 levels rescues the aberrant cardiovascular choices and premature CM differentiation observed upon lamin A/C ablation. These data are consistent with a previous study showing that GATA4 promotes CM and represses the alternative endothelial/endocardial gene expression[32].

We next corroborated our findings in vivo using CMV-Cre mediated germline *Gata4* ablation, to delete one functional *Gata4* allele in wild-type, *Lmna*+/− and *Lmna*−/− mice. Indeed, *Gata4*+/−*Lmna*−/− P1 mice had similar heart weight/body weight (HW/BW) to wild-type mice in contrast to *Gata4*+/+*Lmna*−/− P1 mice that showed significantly higher HW/BW (Supplementary Fig. 9c). Moreover, we observed a pronounced rescue of the decreased capillary density in hearts from both *Gata4*+/−*Lmna*+/− and *Gata4*+/−*Lmna*−/− compared to *Lmna*+/− and *Lmna*−/− (Fig. 9h, i), whereas CMs size in cross sections did not change (Supplementary Fig. 9d). Importantly, we observed a rescue of the reduced CM proliferation and precocious CM binucleation in *Gata4*+/−*Lmna*−/− compared to *Lmna*−/− mice (Fig. 9j, k, Supplementary Fig. 9e), supporting a key role of Gata4 upregulation in cardiac laminopathies.

In summary, this work reveals how disruption of lamin A/C-dependent chromatin architecture in naïve pluripotent stem cells results in a cardiac disease phenotype and pinpoints key molecular determinants at the root of *LMNA* loss-of-function cardiomyopathies.

## Discussion

Our study shows a key role of lamin A/C in keeping CM lineage-specific genes in naïve pluripotent stem cells silent, thus preventing epigenetic priming and aberrant cardiovascular cell fate choices during development. Loss of lamin A/C promoted CM and repressed EC fate, as a result of activation of the key cardiac TF Gata4 and precocious induction of a CM-specific gene expression program, as we demonstrate in this study using a combination of in vitro and in vivo model systems (Fig. 10).

The nuclear lamina consists of two separate classes of lamins, A-type and B-type lamins. It has been long thought that while B-type lamins are expressed throughout development, A-type lamins are expressed only highly after commitment of cells to a particular differentiation pathway. However our data clearly demonstrate that lamin A/C is not only expressed in naïve pluripotent stem cells but also plays essential role in naïve pluripotency maintenance and establishment and in preventing precocious CM differentiation. Interestingly, lamin A/C protein levels dramatically decreased with the exit from pluripotency and were only high again in CMs, suggesting that lack of lamin A/C provides a window of opportunity for stem cells to acquire specific fate.

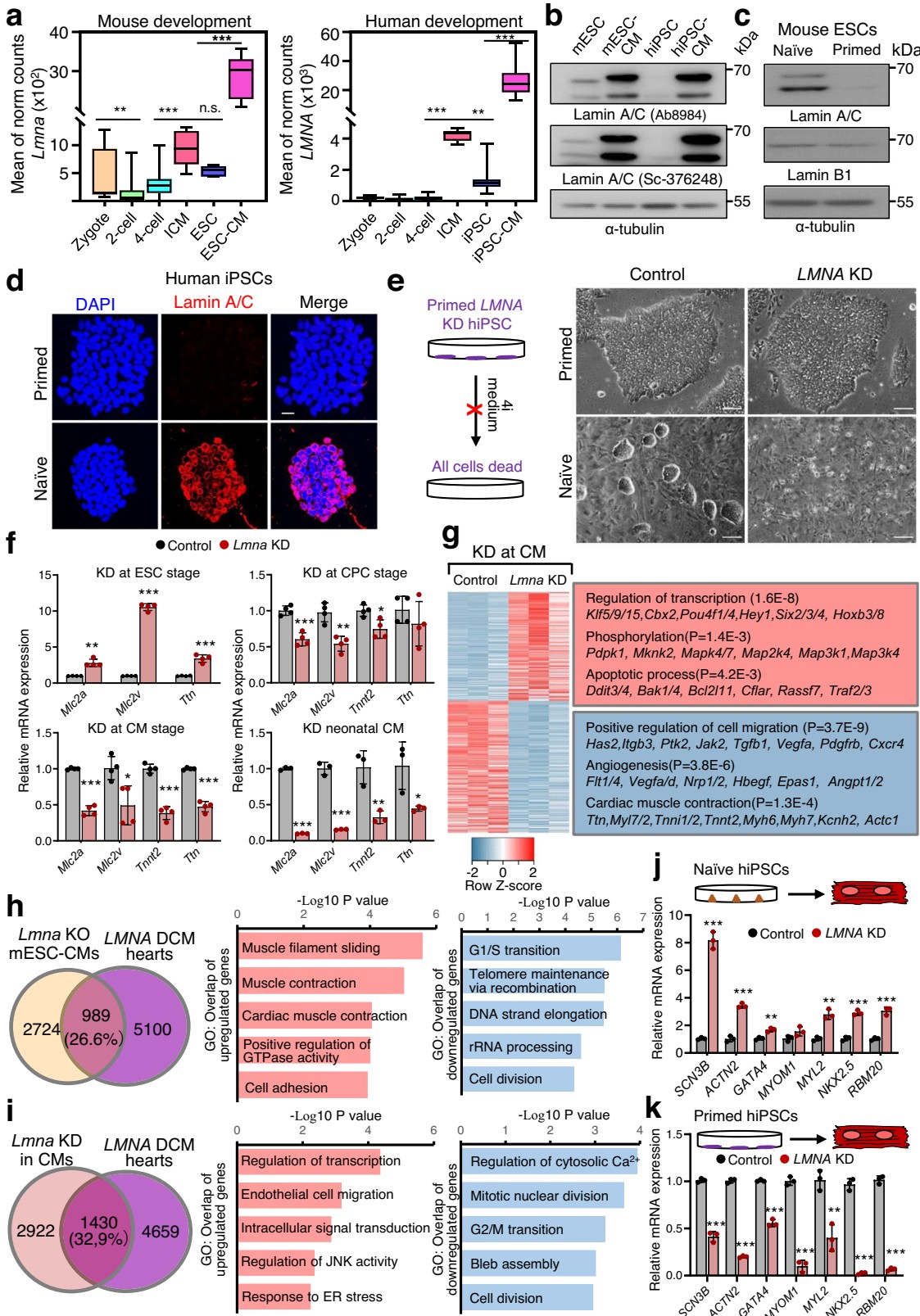

Functionally, A-type and B-type lamins form distinct meshworks[33] and while lamins B1 and B2 are localized at the periphery and associate mainly with transcriptionally inactive chromatin[34,35], lamins A and C are found at the nuclear periphery and in the nuclear interior and associate with both hetero- and euchromatin[36]. However, the loss of A-type lamins results in

alterations in B-type meshworks and vice versa, suggesting that their activity might be interconnected. In mESCs, 92% of lamin B1 LADs overlapped with lamin A LADs, whereas a large numbers of LADs were specific for lamin A/C. Located within LADs specifically associated with lamin A/C were genes for cardiac pioneer TFs, such as Gata4, Gata6 and Isl1[37,38], which might explain why lamin A/C deficiency

**Fig. 5 | Distinct functions of lamin A/C in naïve PSCs and cardiomyocytes for cardiac laminopathies. a** Box plots representing normalized counts for *Lmna* transcripts in RNA-Seq datasets from zygote (*n* = 27), 2-cell (*n* = 42), 4-cell stage (*n* = 60), inner cell mass (ICM, *n* = 9), mESCs (*n* = 6) and mESC-derived CMs (*n* = 6) (GSE57249, left panel). *LMNA* expression in human zygote (*n* = 21), 2-cell (*n* = 49), 4-cell stage (*n* = 84), ICM (*n* = 28), iPSC (*n* = 309) and iPSC-derived CMs (*n* = 91) (GSE36552, GES101571 and GSE107654, right panel). **b** Western blot analysis of lamin A/C in mESCs, mESC-derived CMs, hiPSCs, and hiPSCs-derived CMs using two lamin A/C antibodies. **c** Western blot analysis of lamin A/C in mESCs cultured in naïve and primed conditions. **d** Immunostainings for lamin A/C (red) and DAPI (blue) in hiPSCs cultured in primed and naïve conditions. Scale bars, 20 µm. **e** Phase contrast images of control and *LMNA* knockdown human iPSCs colonies cultured in primed or naïve conditions. Scale bars, 200 µm. **f** Relative expression of CM genes at d10 of directed cardiac differentiation after shRNA mediated *Lmna* knockdown at different stages: mESC (*n* = 4), CP (*n* = 4) and CM stage (*n* = 4), as well as in neonatal CMs (*n* = 3). **g** Heat-map representation of RNA-Seq analysis of CMs after *Lmna* knockdown at CM stage. Representative genes and enriched GO terms are presented on the right. **h**, **i** Overlap of DEG in patients with *LMNA* mutations (GSE120836) and *Lmna*−/− mESCs-derived CMs (**h**) or after *Lmna* knockdown at CM stage (**i**). GO analysis of common upregulated (red) and downregulated (blue) genes are presented in the middle and right panels. **j**, **k** Relative mRNA expression of common upregulated genes in patients with *LMNA* mutations and *Lmna*−/− mESCs-derived CMs in hiPSCs differentiated in CMs after lamin A/C silencing in naïve (**j**, *n* = 3) or primed hiPSCs (**k**, *n* = 3). **g**, **h**, **i** RNA-Seq datasets are from *n* = 3 biologically independent samples. *P*-values are from GO pathway analysis using DAVID Bioinformatics Resources 6.8. **f**, **j**, **k** Values for *n* represent biologically independent samples. Data are mean ± SD; unpaired two-tailed Student's *t*-test was used. P values are as follows: ****p* < 0.001, ***p* < 0.01, **p* < 0.05. Source data are provided as a Source Data file.

results in major defects in cardiovascular differentiation and function, whereas lamin B1 deletion does not show major effect.

Interestingly, we observed major chromatin reorganization at cardiac muscle development and differentiation genes already in naïve pluripotent stem cells. Together with specific changes in chromatin organization, we observed large-scale changes in chromatin compaction and 3D organization of chromosome 14. In line with these findings, major changes were observed in the compaction of chromosome 1, 4, 13, and 14 in lamin B1, B2, and A/C triple KO mouse mESCs[39], suggesting that lamin A might play a specific role in chromosome 14 compaction, whereas lamin B1 and B2 could be involved in compaction of chromosomes 1, 4 and 13.

Further, we observed two modes of lamin A/C-mediated transcriptional control. First, lamin A/C keeps key genes involved in cell differentiation and cardiac morphogenesis silent, such as *Gata4* and *Gata6*, *Bmps*, *Fgf10*, *Wnts*, *Myl4*, etc. Upon lamin A/C depletion these genes are transcriptionally activated as a result of major changes in chromatin localization and organization. Second, lamin A/C restricts transcriptional permissiveness of chromatin at genes involved in cardiac morphogenesis and function, such as *Mef2c*, *Ryr2*, *Mybpc3*, *Lmod2*, *Nebl*, *Adrb2*, etc. Ablation of lamin A/C results in major changes of chromatin localization and structure at these genes resulting in chromatin opening, which, however, is not sufficient to elicit gene activation in ESCs. During cardiac differentiation these epigenetically primed genomic loci are easily accessible to cardiac TFs (e.g., Gata4/6, Hand2, Meis1, Foxo1, etc.), resulting in precocious activation of a gene expression program promoting CM versus EC fate, accompanied by premature CM maturation, binucleation, cell cycle withdrawal and abnormal contractility. In contrast, disease causing *LMNA* point mutations with different clinical manifestations, such as Lmna p.H222P mutation and Lmna p.G609G showed impaired CM differentiation and did not cause dissociation of CM specific genes from the nuclear periphery and similar chromatin opening in pluripotent stem cells, suggesting that the molecular mechanisms resulting in cardiac abnormalities upon *Lmna* haploinsufficiency or changes in lamin A/C protein functionality are distinct.

Gata4 represents a classical example of a pioneer factor, as it can efficiently bind to its target sequences on nucleosomal DNA[37]. Overexpression of Gata4, Tbx5, and the cardiac-specific component of the Brg1 chromatin complex Baf60c reprogramed somatic into cardiac mesoderm[40], and combined expression of Gata4, Tbx5, and Mef2c reprogrammed fibroblasts to a CM-like cell fate[41]. Furthermore, GATA4 has been shown to be a critical regulator of cardiac versus EC fate of cardiovascular progenitors[32] and Gata4 silencing rescued the aberrant cardiovascular cell fate and differentiation of *Lmna*−/− mESCs both in vitro and in vivo, supporting a key role of Gata4 upregulation for cardiac laminopathies. Interestingly, overexpression of Gata4 in mESCs in serum containing medium promoted endoderm but not cardiac mesoderm fate, however overexpressing of any of the cardiac

Gata factors expressed (Gata4/5/6)[42] in serum-free conditions efficiently directed cardiac fate[43], suggesting that growth factor signaling might play an important role in controlling Gata TF functions in cardiogenesis. Thus, the activation of *Bmps*, *Fgf10*, or *Wnts* in lamin A/C-deficient mESCs together with *Gata6* activation might contribute to the precocious activation of the cardiac gene program, which together with premature binding of Meis1 to chromatin can cause an early cell cycle withdrawal of lamin A/C-deficient CMs.

In line with the cell-culture-based phenotype, *Lmna*-deficient and haploinsufficient embryos showed increased expression of pioneer cardiac TFs[38,40] during early cardiogenesis, increased expression of CM-specific structural and contraction genes, hypercontractility as well as precocious binucleation, decreased proliferation and non-compaction cardiomyopathy. Interestingly, we observed more pronounced defects in the RV myocardium, which may be accounted by the different embryological origin of the RV compared to the LV myocardium[44]. The RV is formed by the second heart field (SHF) progenitor cells, marked by the expression of the pioneer TF Isl1[38,45], which is also found in lamin A LADs in ESCs and is significantly upregulated during early cardiogenesis upon lamin A/C loss. Interestingly, in contrast to the first heart field (FHF) progenitor cells, SHF precursors give rise to CMs and ECs[46] and the crosstalk between these two distinct heart cell types is instrumental for proper cardiac development and myocardial compaction[47], suggesting that abnormal cardiovascular cell fate choices and dysfunctional endothelium might pay a role in *LMNA* LOF cardiomyopathy.

Similarly, at birth *Lmna*-null mice showed increased ejection fraction and fractional shortening. While at first sight these findings may appear surprising, as they seem to conflict with the clinical phenotype of patients carrying *LMNA* mutations, our data are consistent with a recent study showing that lamin A/C haploinsufficient hiPSC-CMs are also hypercontractile[17]. Importantly, two weeks after birth heart function was significantly decreased in *Lmna*-deficient and haploinsufficient mice and the left ventricle was dilated, consistent with the clinical phenotype. The sharp contrast to the observations directly after birth is most likely due to the increased cell death specifically in lamin A/C-deficient CMs and to the decreased CM proliferation in the immediate postnatal window (P1-P7), which may be connected to the epigenetic priming at Meis1 and Foxo1 binding sites. Meis1 regulates postnatal CM cell cycle arrest[26] whereas FoxO proteins are key regulators of apoptosis[48] and were shown to be activated and to contribute to the pathogenesis of cardiac laminopathies[27].

In addition, the premature cell cycle arrest and elevated cell death could be due to the inability of *Lmna*−/− CMs to respond adequately to oxidative and mechanical stress. Indeed, mechanosensing by the nuclear lamina have been shown to protect against nuclear rupture, DNA damage, and cell-cycle withdrawal, while the oxygen-rich postnatal environment was proposed to induce cell-cycle arrest through the DNA damage response pathway[31,49]. Interestingly, we observed

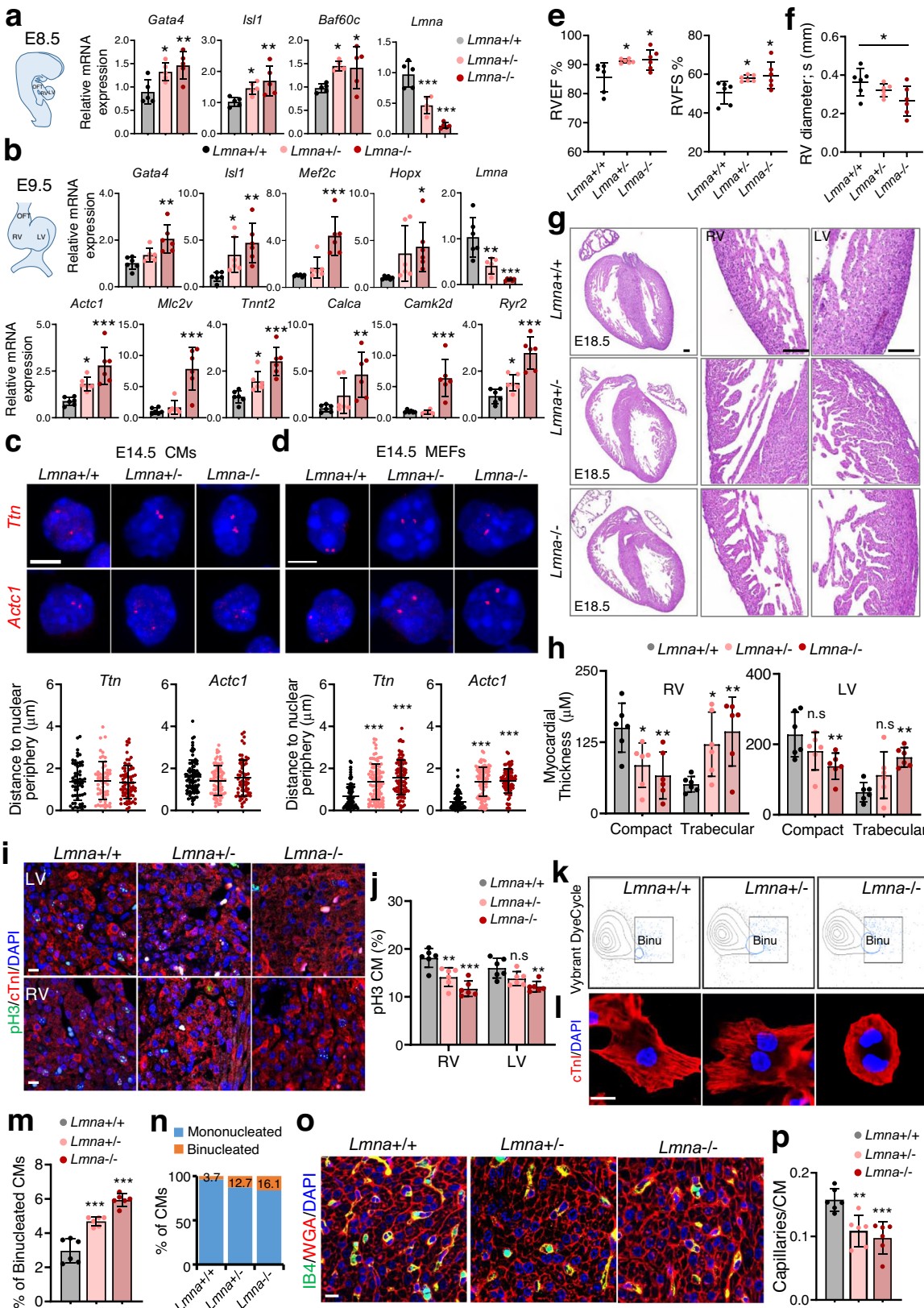

increased DNA damage and cell death in *Lmna−/−* non-CMs subjected to stretch and oxidative stress, suggesting that activation of the DNA damage pathway might contribute to the premature cell cycle arrest and increased cell death upon lamin A/C LOF. Importantly, during embryogenesis the RV and the LV eject blood at a high pressure into the systemic circulation as blood shunts through the ductus arteriosus

and foramen ovale, while after birth the pulmonary circulation is a low-pressure circuit. Thus, the differences observed in RV and LV function during embryogenesis and after birth might be due to the important function of lamin A/C in mechanosensing and response.

In addition to cardiac hyperfunction, vascular rarefaction is also one of the earliest functional changes in *Lmna* knockout-induced

**Fig. 6 | Premature cardiomyocyte binucleation and non-compaction cardio-myopathy upon lamin A/C loss-of-function. a, b** qPCR analysis of dissected pharyngeal mesoderm/ hearts of E8.5 wild-type, *Lmna*+/- and *Lmna*−/− embryos (**a**) as well as dissected hearts of E9.5 embryos (**b**) for cardiac TFs and CM genes. **c, d** Representative DNA FISH images of *Ttn* and *Actc1* gene loci in CMs (**c**, top) and fibroblasts (**d**, top) isolated from E14.5 embryos and quantification of the distance of these loci to the nuclear periphery (bottom). *n* = 30-50 cells were used for quantification. Scale bars, 4 μm. **e, f** Right ventricular ejection fraction (RVEF), right ventricular fractional shortening (RVFS) (**e**) and RV systolic diameter (**f**) assessed by echocardiography in wild-type, *Lmna*+/− and *Lmna*−/− embryos at E16.5. **g** H&E staining of representative heart sections from wild-type, *Lmna*+/− and *Lmna*−/− embryos. Magnified images of indicated right ventricle (RV) and left ventricle (LV) regions are shown in the middle and right panel. Scale bars, 100 μm. **h** RV and LV wall thickness (compact myocardium) and trabecular layer thickness. **i, j** Immunostaining of heart sections for phospho-histone H3 (pH3, green), cardiac

troponin I (cTnI, red) and DAPI (blue) at E18.5 (**i**) and quantification of the per-centage of mitotic RV and LV CMs at E18.5 (**j**). Scale bars, 10 μm. **k, m** Representative FACS plots of cells stained with Vybrant DyeCycle DNA dye (**k**). The % of binu-cleated CMs was determined after removing ECs (**m**). **l, n** Immunofluorescence staining of E18.5 CMs cultured for 12 h after isolation with cTnI (red) and DAPI (blue) (**l**) and quantification of binucleated CMs (**n**). Scale bars, 10 μm. **o, p** Immunofluorescence staining of E18.5 heart sections with isolectin B4 (IB4, green), wheat germ agglutinin (WGA, red) together with DAPI (blue) (**o**) and quantification of capillaries/CM ratio (**p**). Scale bars, 10 μm. **a** *n* = 5 embryos with the indicated genotype; **b–p**: *n* = 6 embryos with the indicated genotype. **a–f, h, j, m, p** Data are mean ± SD; differences between groups were assessed using one-way ANOVA with Tukey correction multiple comparisons. *P*-values are as fol-lows: \*\*\**p* < 0.001, \*\**p* < 0.01, \**p* < 0.05. Source data are provided as a Source Data file.

cardiac laminopathies. To what extent vascular rarefaction and dys-function contribute to the cardiac phenotype will need further inves-tigation. Indeed, recent study have suggested that vascular dysfunction mediated through KLF2 may contribute to the patho-genesis of *LMNA*-related DCM[50].

Importantly, we observed different epigenetic and transcriptional alterations upon lamin A/C depletion in naïve pluripotent stem cells or specifically in CMs. Both of these contributed to the transcriptional changes detected in patients with *LMNA*-associated DCM mutations, suggesting distinct functions of lamin A/C-dependent chromatin architecture in committed versus uncommitted cells for cardiac development and disease. In contrast to human blastocysts and naïve mouse mESCs, hiPSCs cultured in standard conditions represent a primed state and do not express detectable levels of lamin A/C protein[18], suggesting that some important aspects of lamin A/C func-tion could not be modelled using hiPSCs as in previously published studies[16–18] and requires studies using naïve hiPSCs carrying *LMNA* mutations. Since we observed an important role of Lamin A/C in naïve pluripotency establishment and maintenance it would be important to address whether these processes are affected in hiPSCs carrying dif-ferent pathogenic *LMNA* mutations.

In conclusion, our study highlights a central role for lamin A/C in naïve pluripotent stem cells in instructing cardiovascular cell fate and differentiation, provides important insights in the primary events underlying cardiac laminopathies and highlights potential opportu-nities for their treatment.

## Methods
### Mouse lines
The *Lmna* tm1.1Yxz/J line[29] and the *Gata4* tm1.1Sad were obtained from Jackson Laboratory and were maintained on a C57BL/6J background. Both male and female mice at the indicated in the figure legends age were used within the study. Mice were housed in a pathogen-free animal facility under standard conditions with a 12 hour light/dark cycle, temperature of 20−25 degrees, and humidity range of 30−70%. All animal experiments were performed according to the regulations issued by the Committee for Animal Rights Protection of the State of Baden-Württemberg (Regierungspraesidium Karlsruhe, permit num-ber: G-194/18).

### Cell lines and cell culture
HEK293T cells were purchased from ATCC (CRL-3216) and cultured in DMEM supplemented with 10% FCS, 1% penicillin-streptomycin (Thermo Fisher Scientific, 15140122), and 2 mM L-glutamine (Thermo Fisher Scientific, 25030123).

Murine E14-NKX2-5-EmGFP mESCs generated by Hsiao et al.[51] were maintained on mitomycin (Sigma, M4287) treated mouse embryonic fibroblasts (MEF) in DMEM high glucose (Thermo Fisher Scien-tific,10938025) supplemented with 15% fetal bovine serum (FBS,

Thermo Fisher Scientific,10270106), 2 mM L-glutamine (Thermo Fisher Scientific, 25030123), 0.1 mM 2-mercaptoethanol (Sigma, M3148), 0.1 mM non-essential amino acids (Thermo Fisher Scientific, 11140035), 1 mM sodium pyruvate (Gibico,11360070), 4.5 mg/ml D-glucose, 1% penicillin-streptomycin (Thermo Fisher Scientific, 15140122) and 1000 U/ml leukemia inhibitory factor (LIF ESGRO, Mil-lipore, ESG1107).

### Mouse mESC lines generation and differentiation
For generation of *Lmna*−/−, *Lmna*+/−, *Lmna H222P/H222P*, *Lmna G609G/+*, *Lmna G609G/G609G*, *Lmnb1*−/− and *Gata4*+/− murine mESCs by CRISPR/Cas9-mediated gene targeting, a single guide RNA or a combination of two guide RNAs (gRNAs) and mutation donor templates were used as follows: *Lmna* gRNA-1: 5′-CACCGCACTGCT CACGTTCCACCAC-3′ and *Lmna* gRNA-2: 5′-CACCGAGCTATCAGC ACTCTGTTAT-3′; *Lmnb1* gRNA-1: 5′-CACCGAAACTCTAAGGATGCG GCGC-3′ and *Lmnb1* gRNA-2: 5′-CACCGAGAGGCTCTCGATCCTCAT C-3′; *Gata4* gRNA-1: 5′-CACCGTGCAGTCTCCACCGGCTCGT-3′ and *Gata4* gRNA-2: 5′-CACCGCTGTCAGGAGCACGGCTAAT-3, *Lmna* H222P gRNA: 5′- CACCGTGACAAGGCTGCCGGTGGAG-3, *Lmna* H222P donor: 5′- CCT TAA CCC TTT CAG GAG CTC CGT GAG ACC AAG CGC CGG CCT GAG ACG CGG CTT GTG GAG ATC GAT AAC GGA AAG CAG CGA GAG TTT GAG AGC CGG CTGGCAGATGCCCTGCAGGAGCT-3′, *Lmna* G609G gRNA: 5′- CACCGGGCTTGTGGAGATCGATAAC-3, *Lmna* G609G donor:5′-CCGTGCTGTGCGGGACGTGTGGGCAGCCTGCTGACAAGGC TGCCGGTGGAGCAGGAGCCCAGGTGGGTGGATCCATCTCCTCCGGA TCTTCTGCCTCCAGTGTCACAGTCACTCGAAGC-3′. Annealed gRNAs were ligated in pSpCas9(BB)−2a-Puro (PX459) V2.0 plasmid, a kind gift from Feng Zhang (Addgene, 62988). Recombinant plasmids or muta-tion donors were transfected into WT mESCs using Lipofectamine 2000 (Thermo Fisher Scientific, 11668019) according to the manu-facturer's protocol. Positive cells were selected after 24 h of transfec-tion for 48 h using puromycin (Thermo Fisher Scientific, A1113803, 4 μg/ml) and 5 000 cells were further plated on a 6-cm dish with fee-ders. After 7 days of culture, single clones were picked and screened by PCR and enzyme digestion, followed by Sanger sequencing.

For stable knockdown mESC lines generation, HEK293T cells were seeded on a 6-well plate and cultured in DMEM, high glucose, GlutaMAX (Gibco, 61965059) supplemented with 10% fetal bovine serum and 1% penicillin-streptomycin. Cells were transfected at 70% confluence with 1.5 μg *Lmna* (GCTTGACTTCCAGAAGAACAT), *Gata4* (CATCTCCTGT-CACTCAGACAT) or control (pLKO) shRNA obtained from the RNAi consortium (TRC) shRNA library along with 0.975 μg CMVΔR8.74 packaging plasmid and 0.525 μg VGV.G envelope plasmids using the X-tremeGENE DNA transfection reagent (Roche, 6366236001). 48 h after transfection the viral supernatant was collected and 2 ml virus were used to transduce 100 000 mouse mESCs in the presence of 8 μg/ml polybrene (Sigma, TR-1003-G) on 5% poly-HEMA (Sigma, P3932) treated 6-well plates for 12 h. The following day, the transduced mESCs

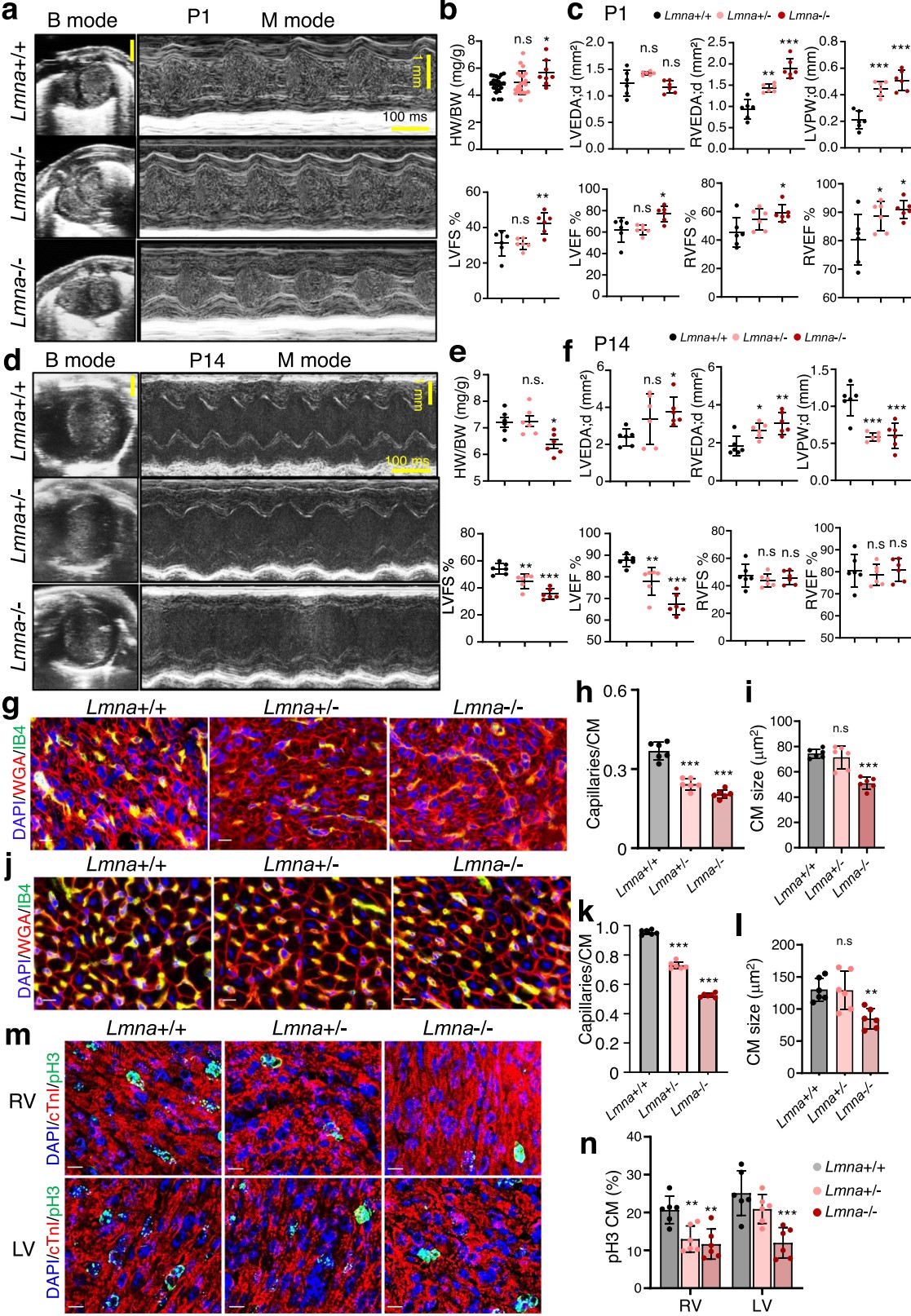

were plated on feeders. 24 h after transduction, the transduced mESCs were selected with 10 ng/mL puromycin for 48 h.

mESCs were differentiated into CMs using either undirected differentiation in hanging droplets or by a directed differentiation method, as described before[38]. Briefly, differentiation through the hanging drop method was initiated following ESCs dissociation by 0.05% trypsin-EDTA (Thermo Fisher Scientific, 25300054) and suspension at $3.3 \times 10^4$ cells/ml in ESC growth medium without LIF in 15 µl drops. After two days, the resulting embryoid bodies (EBs) were transferred to low attachment culture dishes (Greiner Bio-One, 633180). At day 6 (d6) and d10, EBs were dissociated and CPCs and CMs, respectively, were FACS sorted for Nkx2-5-GFP expression. For

**Fig. 7 | Abnormal cardiac composition and function elicited by Lmna deficiency. a, d** Representative examples of B-mode and LV M-mode echocardiograms of wild-type, *Lmna+/-* and *Lmna−/−* mice at P1 (**a**) and at P14 (**d**). **b, e** Heart weight to body weight ratio (HW/BW) in wild-type (*n* = 23), *Lmna+/−* (*n* = 21) and *Lmna−/−* (*n* = 8) mice at P1 (**b**) and at P14 (*n* = 6 for each group) (**e**). **c, f** Left and right ventricular end-diastolic area (LVEDA;d and RVEDA;d), left ventricle posterior wall thickness (LVPW;d), left ventricular fractional shortening (LVFS) and ejection fraction (LVEF) as well as RVFS and RVEF assessed by echocardiography in wild-type (*n* = 6), *Lmna+/−* (*n* = 6) and *Lmna−/−* (*n* = 6) mice at P1 (**c**) and at P14 (*n* = 6 for each group) (**f**). **g, j** Immunofluorescence staining of heart sections for isolectin B4 (IB4, blood vessels, green), wheat germ agglutinin (WGA, CMs, red) together with DAPI (blue) at P1 (**g**) and at P14 (**j**). Scale bars, 10 μm. **h, k** Quantification of the capillaries/CM ratio in wild-type (*n* = 6), *Lmna+/−* (*n* = 6) and *Lmna−/−* (*n* = 6) mice at P1 (**h**) and at P14 (**k**). **i, l** Quantification the CM cross-sectional area in wild-type (*n* = 6), *Lmna + /−* (*n* = 6) and *Lmna−/−* (*n* = 6) mice at P1 (**i**) and at P14 (**l**). **m, n** Immunostaining of heart sections for the mitotic marker phospho-histone H3 (pH3) (green), cTnI (red) and DAPI (blue) (**m**) and quantification of the percentage of mitotic right and left ventricle (RV and LV) CMs (**n**), in wild-type (*n* = 6), *Lmna+/−* (*n* = 6) and *Lmna−/−* (*n* = 6) mice at P1. Scale bars, 10 μm. **b, c, e, f, h, i, k, l, n** Data are mean ± SD; differences between groups were assessed using one-way ANOVA with Tukey correction multiple comparisons. *P*-values are as follows: \*\*\**p* < 0.001, \*\**p* < 0.01, \**p* < 0.05. Source data are provided as a Source Data file.

directed cardiomyocyte differentiation, ESCs were maintained on mouse feeders in Knockout DMEM medium (Thermo Fisher Scientific, 10829018) containing 4.5 mg/ml D-glucose, supplemented with 10% serum replacement (Thermo Fisher Scientific, 10828028), 2 mM L-glutamine, 0.1 mM 2-mercaptoethanol, 1 mM sodium pyruvate, 1% penicillin-streptomycin and 1000 U/ml leukemia inhibitory factor. Before differentiation, ESCs were dissociated and grown on 0.1% gelatin (Sigma, G9391) coated 10-cm dishes in Neurobasal medium: DMEM/F12 (1:1; Thermo Fisher Scientific, 21103049 and 21331020) supplemented with 2000 U/ml LIF and 10 ng/ml BMP4 (R&D, 314-BP) for 2 days. Differentiation was initiated by aggregation in low attachment bacterial dishes at a cell density of 75 000 cells/ml in IMDM: F12 medium (3:1; Thermo Fisher Scientific, 12440053 and 11765054). After 48 h aggregates were dissociated and re-aggregated in the presence of Activin A (R&D, 338-AC, 5 ng/ml), VEGF (R&D, 293-VE-010, 5 ng/ml), and BMP4 (R&D, 314-BP, 0.1-0.8 ng/ml, BMP4 concentration was optimized for each lot). 40 h following the second aggregation, aggregates were dissociated and plated as a monolayer in Stempro-34 medium (Thermo Fisher Scientific, 10639011), supplemented with 2 mM L-glutamine, L-ascorbic acid (Sigma, A4403, 50 μg/ml), VEGF (R&D, 293-VE-010, 5 ng/ml), bFGF (R&D, 233-FB, 10 ng/ml), and FGF10 (R&D, 345-FG, 25 ng/ml) growth factors. Analysis were done at day 5 (CPC stage) and at day 10 (CM stage).

### Knockdown of *Lmna* at the CP and CM stages
For *Lmna* knockdown at the CP stage, virus particles in Stempro-34 medium supplemented with growth factors were used to transduce cells on 0.1% gelatin-coated 24-well plates 7 h after the second aggregation of the direct cardiac differentiation protocol in the presence of 8 μg/ml polybrene. 24 h after transduction, the medium was replaced with normal differentiation medium. CMs at d10 were harvested for analysis. For *Lmna* knockdown at the CM stage, d8 monolayer CMs were trypsinized and 50% of the cells were re-seeded on 0.1% gelatin-coated 24-well plates in the presence of 8 μg/ml polybrene together with virus produced in Stempro-34 medium. The medium was changed to normal Stempro-34 medium 24 h after virus transduction and CMs were harvested at d10.

### siRNA-mediated gene knockdown
siRNA-mediated knockdown experiments were performed as follows: 30 000 mESCs were seeded in 6-well plates with feeders. 24 h later, cells were transfected with either 50 nM scrambled siRNA or 50 nM siRNA against *Gata4* (Dharmacon/GE, L-040759-01-0005), *Lmna* (5′-GAGAGCAGGCCUGAAGCCAAAGAAA-3′), *Lmnc* (5′-TCTCCCACCTCCATGCCAAAG-3′) using Lipofectamine RNAiMax (Thermo Fisher Scientific, 13778-075). 48 h after transfection, the cells were subjected to hanging drop differentiation or further analysis.

### Primed hiPSC cell culture, differentiation, and conversion to naïve hiPSC
The use of hiPSC within this study has been approved by the Ethics Committee of the Medical Faculty Mannheim, University of Heidelberg (approval number: 2022-539). Human iPSC line were cultured in a primed state on Matrigel-coated plates in Essential 8 (E8) medium (Thermo Fisher Scientific). Cells were passaged as small clumps with EDTA, and 5 μM ROCK inhibitor Y-27632 (Miltenyi Biotec) was added for the first 24 h. For hiPSCs differentiation, cells were seeded into 24-well plate precoated with Matrigel and allowed to reach 85–95% confluence (day 0). On day 0, medium was changed to cardiac differentiation medium (RPMI-1640 with glutamine supplemented with 500 μg/ml BSA and 200 μg/ml ascorbic acid) and supplemented with 6 μM CHIR99021. At day 2, medium was substituted with cardiac differentiation medium supplemented with 5 μM IWP2, followed by a change to cardiac differentiation medium on day 4. From day 6 on, cells were cultured in RPMI-B27 media (RPMI with 1× B-27 supplement).

For naïve hiPSCs differentiation, cells were dissociated by 0.05% trypsin-EDTA and cultured in hanging drops, with a density of 500 cells per drop in medium (DMEM medium with 15% FCS, 1% nonessential amino acids, 0.1 mM β-mercaptoethanol, 1%Penicillin-Streptomycin), in the absence of LIF. After two days, the resulting embryoid bodies were transferred to low attachment culture dishes and cultured in differentiation medium. Day10 EBs were harvested for analysis. For transfection, hiPSCs were cultured feeder free with 5 μM ROCK inhibitor for 24 h before virus infection. After infection, cells were subsequently plated on Matrigel coated plate or MEF feeder layers.

Conversion of primed human iPSC line to naïve state was performed as described before[52]. Briefly, primed human iPSCs were seeded on feeder cells as small clumps in E8 medium supplemented with 5 μM ROCK inhibitor. 24 h later, medium was switched to 4i medium (knockout DMEM, 1% AlbuMAXI, 1 X N2 supplement, 12.5 μg/ml recombinant human insulin, 20 ng/ml of recombinant human LIF, 8 ng/ml recombinant bFGF and 5 ng/ml recombinant IGF, 1 mM glutamine, 1% nonessential amino acids, 0.1 mM β-mercaptoethanol, 1% Penicillin-Streptomycin and small molecule inhibitors: 1 μM PD0325901; 3 μM CHIR99021; 10 μM SP600125 and 2 μM BIRB796). Following an initial wave of widespread cell death, dome-shaped naïve colonies appeared about 6 to 8 days and were passaged by accutase tratement for 3–5 min.

Toggling of naïve mESCs to primed state was performed as described before[53]. Briefly, naïve mESC were preplated and seeded on 0.1% gelatin coated plate in DMEM/F12 supplemented with 20% knockout serum replacement, 1 mM sodium pyruvate, 0.1 mM nonessential amino acids, 0.1 mM β-mercaptoethanol,1% Penicillin-Streptomycin, 10 ng/ml Activin A, and 10 ng/ml bFGF.

### Neonatal cardiomyocytes isolation and culture
Neonatal CMs were isolated from 1-day-old (P1) mice using a neonatal heart dissociation kit (Miltenyi Biotec, 130-098-373) and neonatal mouse CM isolation kit (Miltenyi Biotec, 130-100-825), according to the manufacturer's instructions. The cells were then cultured on coverslips in 24-well plates or glass bottom microwell dishes pre-coated with 10 μg/ml Laminin (Corning, 354232) in DMEM/F12 medium supplemented with 1% L-glutamine, 1% Na-pyruvate,1 % non-essential amino acids, 1% penicillin/streptomycin, 5% horse serum and 10% FBS.

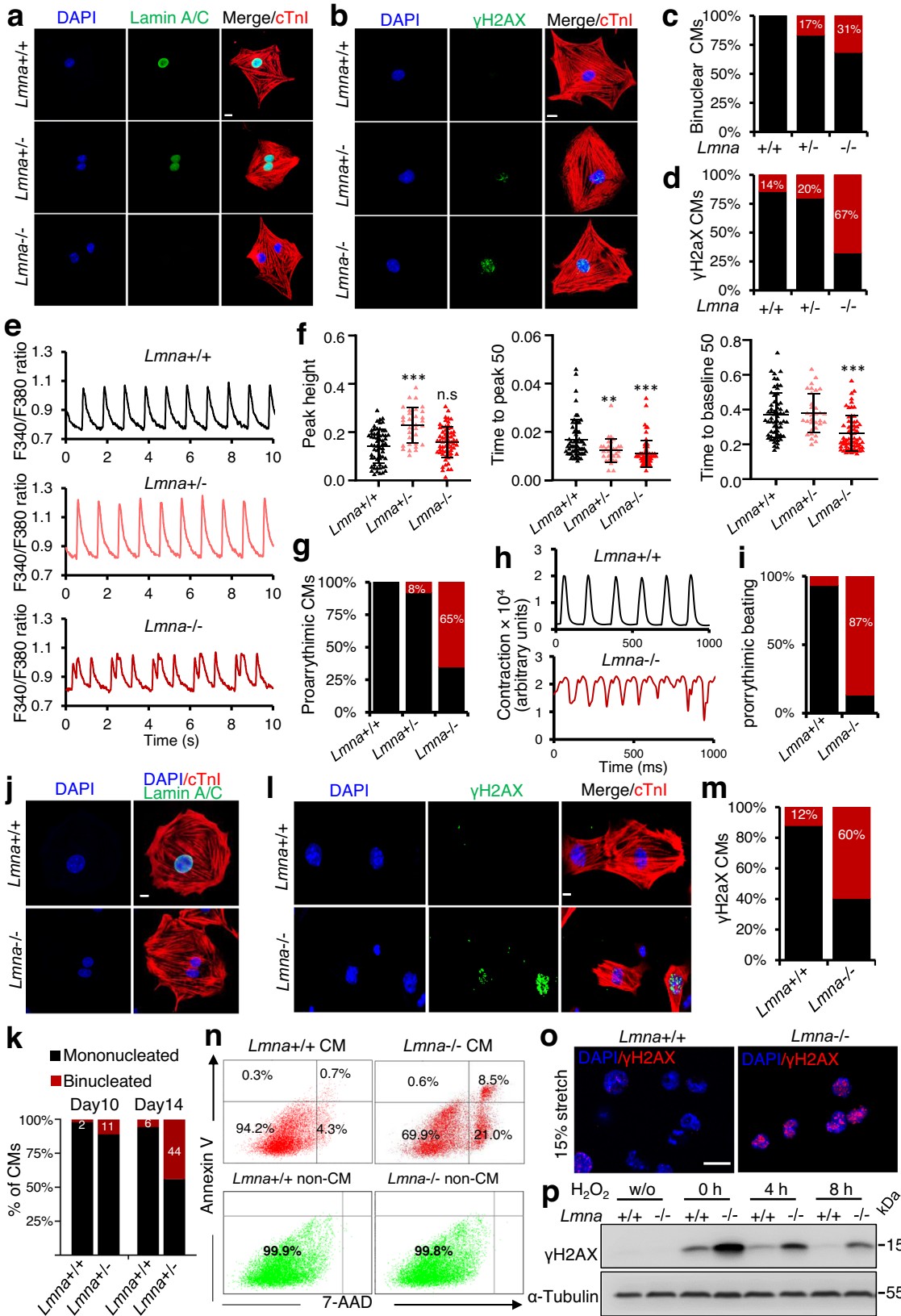

## Measurements of calcium transients in ventricular CMs

50 000 freshly isolated neonatal CMs from *Lmna+/+*, *Lmna+/-* and *Lmna−/−* mice were seeded on glass bottom microwell dishes (MatTeK, P35g-1.5-20-C) pre-coated with 10 μg/ml laminin (Corning, 354232). After 2 days, CMs were washed twice with Tyrode's solution (140 mM NaCl, 5.4 mM KCl, 1 mM MgCl$_2$, 25 mM HEPES, 1 mM CaCl$_2$, 10 mM glucose, pH7.4) and then loaded with 2 μM Fura-2, AM (Thermo Fisher Scientific Fisher, F1221) in Tyrode's solution at 37 °C for 20 min. The cells were then washed three times for 5 min with Tyrode's solution at 37 °C. Calcium transients were measured at 10 V and 1 Hz stimulation on MyoPacer (IonOptix) and data were analyzed with IonWizard 7.4. Fura-2, AM-loaded cells were excited at

**Fig. 8 | Loss of lamin A/C induces precocious CM binucleation, cell death and a proarrhythmic phenotype. a, b** Representative immunostaining of CMs isolated from *Lmna*+/+ (*n* = 6), *Lmna*+/− (*n* = 15) and *Lmna*−/− (*n* = 8) mice at P1 for lamin A/C (green), cTnI (red) and DAPI (blue) (**a**) or for γH2AX (green), cTnI (red) and DAPI (blue) (**b**). **c, d** Quantification of binucleated CMs (**c**) and of γH2AX positive CMs (**d**). **e** Representative Ca2+ transients traces recorded using fura-2-labeled P1 *Lmna*+/+, *Lmna*+/− and *Lmna*−/− CMs. The ratio of the fura-2 AM signal excited at 340 nm and 380 nm (F340/F380) is shown. **f** Peak height, time to peak 50% and time to baseline 50% of Ca2+ traces of *Lmna*+/+, *Lmna*+/− and *Lmna*−/− CMs isolated from P1 mice. 35–60 cells from three independent CM isolations were measured. **g** Percentage of CMs exhibiting pro-arrhythmic Ca2+ waves. **h, i** Plot of contraction amplitude and speed in spontaneously beating CMs at d10 extracted from image sequences using MUSCLEMOTION V1.0 (**h**) and quantification of proarrythmic beating (**i**).

**j, k** Immunofluorescent staining of FACS-sorted Nkx2-5+ CMs from d10 EBs for lamin A/C (green), cTnI (red) and DAPI (blue) (**j**), and quantification of binucleated CMs at d10 and d14 (**k**). **l, m** Immunostaining of FACS-sorted Nkx2-5+ CMs from d10 EBs for γH2AX (green), cTnI (red) and DAPI (blue) (**l**) and quantification of γH2AX positive CMs (**m**). **n** Representative FACS analysis of CMs and non-CMs (*n* = 3) in d10 EBs stained with Annexin V-APC and 7-AAD. **o** Immunostaining for γH2AX (red) and DAPI (blue) of *Lmna*+/+ and *Lmna*−/− mESCs subjected to mechanical stretch (15% elongation). **p** Western blot analysis for γH2AX of extracts from *Lmna*+/+ and *Lmna*−/− mESCs either not treated (w/o) or treated with H$_2$O$_2$ (0 h) and allowed to recover in fresh medium for 4 h and 8 h. Scale bars, 10 μm (**a, b, j, l,** and **o**). **f** Data are mean ± SD; Differences between groups were assessed using one-way ANOVA with Tukey correction multiple comparisons. P-values are as follows: ***$p < 0.001$, **$p < 0.01$, *$p < 0.05$. Source data are provided as a Source Data file.

both 340 and 380 nm, and the fluorescence emission signal was collected at 510 nm. Intracellular calcium changes were expressed as changes in the ratio R = F340/F380. For measuring neonatal CMs calcium transients after knockdown of *Lmna*, lentiviruses carrying shRNA for lamin A/C were produced as above and 1 ml supernatant containing viral particles was used to transduce 50 000 neonatal CMs. Calcium transients were measured as above 48 h after transduction.

### Histology

Embryos were sacrificed by cervical dislocation, hearts were dissected and fixed with 4% paraformaldehyde overnight at 4 °C. The hearts were then dehydrated, embedded, and sectioned. H&E staining was performed according to the manufacturer's instruction (GHS116, HT-110216; Sigma-Aldrich). Representative images of histological analysis of mice with the same genotype are presented. Heart sections were imaged by Zeiss Axio Scan (Zeiss).

### Immunofluorescence staining

For immunofluorescence staining on heart sections, 7 μm cryosections or paraffin sections were fixed in 4% paraformaldehyde (Arcos, 416780010) for 20 min, washed three times with PBS, and permeabilized with 0.3% Triton X-100/PBS for 30 min. Slides were then blocked in 3% BSA (Roth, 8076.2) for 30 min and then incubated with anti-phospho-histone H3 (Ser10) (Millipore, 06-570, 1:100), anti-cardiac troponin I (Abcam, ab56357, 1:100); anti-Aurora B (Abcam, ab2254, 1:100) in 1% BSA overnight at 4 °C. On the following day, the sections were washed three times with PBS and incubated with a corresponding secondary antibody, conjugated to Alexa 555 or Alexa 488 (Thermo Fisher Scientific, 1:500) for 2 h. The slides were then washed three times with PBS, each time 5 min, and stained with DAPI in PBS for 15 min at room temperature. Slides were washed with PBS three times and mounted with Mowiol 4–88 (Millipore, 475904) mounting medium. For paraffin sections, rehydration and antigen retrieval were conducted before staining.

For immunofluorescence staining cells were permeabilized for 10 min with 0.5% Triton X-100 in PBS. After blocking in 3% BSA in PBS for 30 min, mESCs were incubated with anti-lamin A/C (E-1, Santa Cruz, sc376248, 1:100), anti-lamin A/C (131c3, Abcam, ab8984, 1:100), anti-lamin B1 (Abcam, ab16048, 1:100), Anti-lamin B1 (Sigma, HPA050524, 1:100), anti-Gata4 (C-20, Santa Cruz, sc1237, 1:100), anti-Myl4 (Sigma, HPA051884, 1:100) and anti-Oct3/4 (Santa Cruz, sc5279, 1:100) antibodies in 0.3% Triton X-100/ 1% BSA/ 1xPBS overnight in a humidified chamber at 4 °C. After three consecutive 5 min washes in PBS, mESCs were incubated for a further 1 hour with a corresponding secondary antibody, conjugated to Alexa 555 or Alexa 488 in PBS followed by DAPI staining. Antibodies used in this study are listed in Supplementary Data 6.

### Wheat germ agglutinin (WGA) and isolectin B4 (IB4) co-staining

For isolectin B4 (IB4) and wheat germ agglutinin (WGA) staining, 7 μm cryosections were fixed in 4% paraformaldehyde for 20 min, washed three times with PBS, blocked in 3% BSA for 30 min and then incubated with WGA Alexa 488 Conjugate (Thermo Fisher Scientific, W11261, 1 mg/ml, 1:100) antibody diluted in 1% BSA for 1 hour at room temperature in a humidified dark chamber. Slides were then washed with PBS three times, each time 5 min, permeabilized with 0.3% Triton in PBS for 30 min, and then incubated with IB4 Alexa 568 conjugate (Thermo Fisher Scientific, I21412, 1 mg/ml, 1:100) diluted in 1% BSA for 2 h at room temperature in a humidified dark chamber. The slides were then washed three times with PBS for 5 min and stained with DAPI in PBS for 15 min at room temperature. After DAPI staining, slides were washed three times with PBS and mounted with Mowiol 4–88 mounting medium.

### DNA Fluorescence In Situ Hybridization (FISH)

DNA FISH probes for *Gata4*, *Mef2c*, *Actc1*, *Ttn*, and *Kcnq1* were labeled with digoxigenin by Nick Translation kit (Roche,10976776001) according to the manufacturer's protocol using BAC DNA clones: *Gata4* (BACPAC Resources, RP23-124M15), *Mef2c* (BACPAC Resources, RP23-187h18), *Actc1* (BACPAC Resources, RP23-196j13), *Ttn* (BACPAC Resources, RP23-310F9), *Kcnq1* (BACPAC Resources, RP23-207g7). For DNA FISH, 60000 mESCs were attached on poly-L-lysine-coated coverslips for 10 min at 37 °C. Cells were washed with PBS, three times, and fixed in 4% paraformaldehyde (Electron Microscopy Sciences, 15710) at room temperature for 10 min. Afterwards, cells were rinsed in PBS, permeabilized with 0.5% Triton X-100/PBS for 10 min at room temperature and rinsed briefly in PBS. Cells were then incubated in 20% glycerol (Roth, 3783.1) in PBS for a minimum of 60 min at room temperature and were left at 4 °C overnight. The next day, the coverslips were removed from the glycerol solution, dipped into liquid nitrogen for about 15 s until the coverslips were completely frozen and thawed at room temperature. The freeze/thaw treatment was repeated 6 times. The cells were then washed with 0.05% Triton X-100/PBS, three times, for 5 min. After a brief wash with 0.1 N HCl the cells were incubated in a fresh 0.1 N HCl for 10 min followed by three washes with 0.05% Triton X-100/PBS, 5 min each, one wash in 2xSSC (0.3 M NaCl, 30 mM sodium citrate) and prehybridized in 50% formamide (Sigma, F9037)/2xSSC for 4 h at 37 °C. 5 μl labeled DNA probes in hybridization buffer were added to the slides and sealed with rubber cement. Slides were then incubated at 75 °C for 3 min and hybridized for 2 days at 37 °C. Hybridization was followed by three washes with 2xSSC, 3x10min at 37 °C and two washes with 0.1xSSC at 60 °C. Afterwards, cells were incubated with anti-digoxigenin-rhodamine, Fab fragments (Roche, 11207750910, 1:200) in 4% BSA/4xSSCT for 1 h at room temperature followed by two washes with 4xSSCT at 37 °C, a brief wash in PBS and post-fixation in 2% formaldehyde/PBS for 10 min. Cell nuclei were counterstained with DAPI/PBS (1:500) for 10 min and after a brief wash

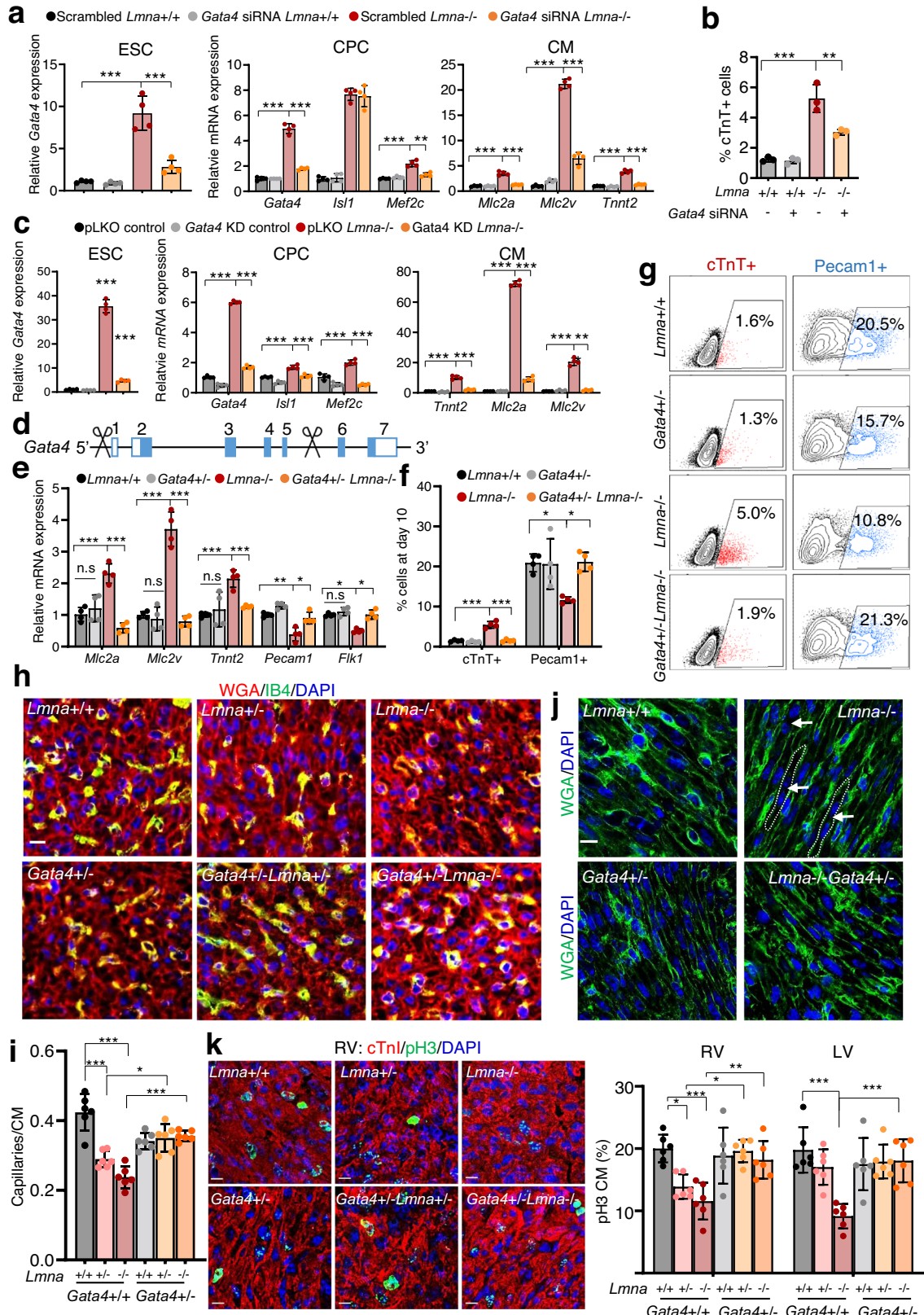

in PBS were mounted with ProLong Diamond Antifade Mountant (Life Technologies, P36930).

### Chromosome painting

Cells were prepared as described for FISH until the permeabilization step. 10 µl of whole chromosome 14 painting probes (MetaSystems, D-1414-050-FI) were applied on the coverslip of a slide followed by sealing with rubber cement for 30 min at room temperature. Slides were then denatured at 75 °C for 2 min and hybridized at 37 °C overnight. Cells were washed with 0.4xSSC (pH 7.0) at 72 °C for 2 min and 2xSSC, 0.05%Tween-20 (pH 7.0) at room temperature for 30 s. Cells were rinsed briefly in distilled water to avoid crystal formation

**Fig. 9 | Gata4 is a crucial mediator of the aberrant cardiovascular cell fate and differentiation elicited by lamin A/C loss. a** Relative mRNA expression of *Gata4* in mESCs (left), cardiac progenitor marker genes in day 6 EBs (CP stage, middle), and CM marker genes in day 10 EBs (CM stage, right) after transient knockdown of *Gata4* by siRNA at the mESC stage (*n* = 4). **b** Percentage of cTnT+ CMs measured by flow cytometry at day 10 of differentiation after transient knockdown of *Gata4* by siRNA in mESCs (*n* = 3). **c** Relative mRNA expression of *Gata4* in mESCs (left), cardiac progenitor marker genes in d6 EBs (CP stage, middle), and CM marker genes in day 10 EBs (CM stage, right) after stable knockdown of *Gata4* by shRNA in mESCs (*n* = 4). **d** Schematic diagram of *Gata4*+/- mESC generation by CRISPR/Cas9-mediated gene editing. **e** Relative mRNA expression of CM (*Mlc2a*, *Mlc2v*, and *Tnnt2*) and EC (*Pecam1* and *Flk1*) marker genes in day 10 EBs (*n* = 4). **f** Percentage of cTnT+ CMs and Pecam1+ ECs measured by flow cytometry at day 10 (*n* = 4). **g** Representative

FACS plots of cTnT+ CMs and Pecam1+ ECs in d10 EBs. **h** Immunofluorescence staining of heart sections for isolectin B4 (IB4, green), wheat germ agglutinin (WGA, red) together with DAPI (blue) at P1. Scale bars, 10 μm. **i** Quantification of the capillaries/CM ratio. **j** Immunostaining of heart sections with wheat germ agglutinin (WGA, red) and DAPI (nucleus, blue). Binucleated CMs are indicated with arrows. Scale bars,10 μm. **k** Immunostaining of heart sections for the mitotic marker phospho-histone H3 (pH3) (green), cardiac troponin I (red) and nucleus (DAPI, blue) at P1 (left) and quantification of the percentage of mitotic RV and LV CMs at P1 (right). Scale bars, 10 μm. **h–k:** *n* = 6 neonatal mice with the indicated genotype. Data are mean ± SD; Differences between groups were assessed using one-way ANOVA with Tukey correction for multiple comparisons. *P*-values are as follows: ***$p < 0.001$, **$p < 0.01$, *$p < 0.05$. Source data are provided as a Source Data file.

**Fig. 10 | Key function of lamin A/C in naïve pluripotent stem cells for cardiac development and disease.** In naïve pluripotent stem cells, lamin A/C tethers cardiac-specific genes and genes involved in stem cell differentiation to the repressive nuclear periphery. Lamin A/C loss leads to their dissociation from the nuclear lamina accompanied by major chromatin reorganization resulting in either ectopic expression in mESCs (e.g., *Gata4*, *Myl4*) or epigenetic priming for activation later in development. These changes lead to precocious activation of a gene expression program promoting CM versus endothelial cell fate, accompanied by premature cardiomyocyte differentiation, cell cycle withdrawal, and abnormal contractility, which is dependent on *Gata4*. The figure was designed with BioRender.

and counterstained with DAPI/PBS (1:500) for 10 min followed by a wash with PBS prior to mounting in ProLong Diamond Antifade Mountant.

**Flow cytometry**
For FACS staining of extracellular markers, EBs were washed with HBSS twice and dissociated into single cells by incubation with 1 mg/ml collagenase I (Cell Systems, LS004196) at 37 °C for 30 min. Cells were then washed with 5% FCS/PBS and blocked in 10% FCS/PBS buffer for 30 min at room temperature. 200 000 cells were used for staining with 2.5 μl APC-conjugated anti-Flk1 (e-Bioscience, 17-5821-81,1:40), PE-conjugated anti-PDGFRα (e-Bioscience, 12-1401-81,1:40), APC-

conjugated anti-Pecam1 (Thermo Fisher Scientific, 17-0311-82,1:40) antibody in 100 μl FACS buffer (0.4% BSA) for 1 hour at room temperature in a dark place. After washing with FACS buffer twice, cells were resuspended in 300 μl FACS buffer and subjected to analysis by BD FACSDiva Software (version8.0.1, firmware version 1.49 BD FACSCanto II).

For FACS staining of intracellular cardiac Troponin T, single cells were prepared as above. After washing with 5% FCS/PBS, 400 000 cells were fixed in 500 μl 3.7% PFA for 30 min at RT. Cells were then washed with 5% FCS/PBS buffer and incubated in permeabilization buffer (0.5% saponin/5% FCS/PBS) for 15 min on ice. Cells were further stained with 2.5 μl APC-conjugated anti-troponin T antibody (BD, 1:40) in 100 μl

permeabilization buffer for 2 h at RT in a dark place, followed by washes with permeabilization buffer, PBS, and FACS buffer and resuspended in 300 µl FACS buffer for FACS analysis.

For apoptosis assay by FACS, one million cells were used for staining with 5 µl of APC Annexin V and 5 µl of 7-AAD in 100 µl Annexin V binding buffer for 15 min at room temperature according to the APC Annexin V Apoptosis Detection Kit with 7-AAD (Biolegend, 640930) instructions. Cells were then resuspended in 400 µl of Annexin V Binding Buffer and subjected to FACS analysis. CMs and non-CMs were distinguished by Nkx2-5 GFP expression.

To assess CM binucleation by FACS, E14.5 and E18.5 hearts were freshly harvested, washed with cold HBSS and dissociated into single cells by incubation with digestion buffer (1 mg/ml collagenase I and 1 mg/ml DNase I in HBSS) at 37 °C 30 min. Cell suspension was then applied to a 70 µm cell strainer to remove cell debris. Single cells were stained with APC-conjugated anti-Pecam1 antibody for 30 min at 4 °C followed by washing with cold HBSS/10%FCS three times. Cells were then stained with Vybrant™ DyeCycle™ Green (Thermo Fisher Scientific, 1:500 dilutions by HBSS/10%FCS) at 37 °C for 30 min and were subjected to FACS analysis.

## Western blotting
Cells were collected and lysed in RIPA buffer supplemented with protease inhibitor cocktail (Millipore, 535142). Protein concentration was quantified using the Pierce BCA protein assay kit (Pierce Biotechnology, 23225). After separation via SDS–PAGE, proteins were transferred to nitrocellulose membranes, blocked in 5% skimmed milk/PBST, and incubated with appropriate primary antibodies: lamin A/C (Abcam, ab8984; 1:1000), lamin A/C (Santa Cruz, sc-376248; 1:1000), lamin B1 (Abcam, ab16048; 1:1000), H3k9me3 (Abcam, ab8898; 1:5000), H3 (Abcam, ab1791; 1:5000), Ryr2 (Sigma, HPA020028, 1:1000), α-tubulin (Sigma, T5168, 1:1000) followed by incubation with a corresponding secondary antibody. The images were acquired on Amersham Imager 600.

## Echocardiographic assessment of cardiac dimensions and function
Transthoracic echocardiography was performed using a Vevo 3100 high-resolution system (Visualsonics, Toronto, ON, Canada) and a 50-MHz MX-700 transducer. Two-dimensional B-mode tracings were recorded in both parasternal long and short axis views at the level of the papillary muscles, followed by one-dimensional M-mode tracings in both axes, wherein at least three consecutive cardiac cycles were used for analysis. Left- and right-ventricular area at end-diastole (LVEDA;d and RVEDA;d, respectively) as well as left-ventricular posterior wall thickness at end-diastole (LVPW;d), were used to characterize cardiac microanatomy while the change of left-ventricular diameter length from end-diastole to end-systole was used to judge cardiac contractility and calculate left-ventricular fractional shortening (FS) and ejection fraction (EF) with Vevo LAB Software Package V3.2.6.

For fetal echocardiography, pregnant mice were anasthetized and one-dimensional AM-mode tracings of E16.5 mouse embryonic hearts were recorded with a 50 MHz MX700 transducer. Right and left ventricular diameter at end-systole, right ventricular diameter at end-diastole, right and left ventricular ejection fraction and fractional shortening (RV diameter;s and LV diameter;s, RV diameter;d, RVEF, LVEF, RVFS and LVFS, respectively) were calculated with the integrated cardiac measurement package, where at least three consecutive cardiac cycles were used for analysis.

## Contraction analysis
EBs at d6 were seeded on 0.1% gelatin-coated 24-well plates. Movies of spontaneous beating at day 8 and day 10 were acquired for at least 30 s

with a Leica DMI8 microscope at 100 frames/sec and a 20× objective. Contractility parameters were obtained by analyzing movies with MUSCLEMOTION V1.0[54].

## RNA Isolation, RT-PCR, and Real-Time PCR
RNA was isolated using the TRIzol RNA Isolation Reagent (Invitrogen, 15596018). For real-time PCR analysis cDNA was synthesized with the High Capacity cDNA Reverse Transcription Kit (Applied Biosystems, 4368813) and real-time PCR was performed using the SYBR GREEN PCR master mix (Applied Biosystems, A25742). Cycle numbers were normalized to these of α-Tubulin (Tuba1a).

## RNA-Sequencing and data analysis
Control and *Lmna* KO Nkx2-5-GFP ES cell lines were differentiated using the hanging drop method. Cardiac progenitors at d6 and CMs at d10 were sorted by Nkx2-5-GFP from three different control and three different *Lmna-/-* clones. Day10 EBs containing CMs, endothelial and other types of cells were also harvested for RNA-Seq. For RNA-Seq of control, *Lmna* knockdown CMs, wild type Nkx2-5-GFP ES cell lines were differentiated by the directed cardiac differentiation method, and CMs were transduced with pLKO control or *Lmna* shRNA lentivirus at d8 and harvested at d10. RNA was isolated using the RNeasy Plus Universal Mini kit (Qiagen #73404). The integrity of the RNA was assessed on a Bioanalyzer 2100 (Agilent). Library preparation and sequencing were performed on a BGISEQ-500 platform. RNA-Seq reads were trimmed of adapters using Trimmomatic-0.39 and mapped to the mm10 reference genome using STAR (−alignIntronMin 20 −alignIntronMax 500000). Read quality was controlled by the MultiQC tool. Reads were counted using the analyzeRepeats.pl function (rna mm10 −count exons −strand both −noadj) from HOMER after creating the tag directories with makeTagDirectory. For visualization of RNA-seq reads in the genome browser IGV, bam files of the three individual replicates were first merged by BamTools, then the BamCoverage function of deepTools with normalization to RPKM was used to generate the bigwig files (-bs 20−smoothLength 40−normalizeUsing RPKM −e 150). Differential expression was quantified and normalized using DESeq2. Reads per kilobase per millions mapped (RPKM) was determined using rpkm.default from EdgeR. Excel was used to filter differentially regulated genes (fold change ≥1.5; log2 fold change ≤ −0.58, ≥0.58; p-value< 0.05.). Gene ontology pathway analysis was performed using DAVID Bioinformatics Resources 6.8. Clustering analysis displayed as heatmap was performed using heatmapper.ca. All Heatmaps represent the row-based Z-scores calculated from trimmed mean of M-values (TMM). The PCA plots were obtained using prcomp into a custom R-script and volcano plots were obtained using a custom R-script. Boxplots were performed with the function boxplot from library Graphic from R using RPKM normalized values. All data, including publicly available data, were normalized with the same parameters. Programs and algorithms used in this study are listed in Supplementary Data 7.

## ATAC-Sequencing and data analysis
*Lmna+/+, Lmna−/−, Lmna H222P/H222P, Lmna G609G/+ and Lmna G609G/G609G mESC and Lmna+/+, Lmna−/− Nkx2-5*-GFP sorted CM were processed for ATAC-Sequencing using the protocol described in[55]. ATAC-Seq reads were trimmed of adapters using Trimmomatic-0.39 and then mapped to mm10 mouse genome with Bowtie2[56]. Read quality was controlled by the MultiQC tool. Following removal of unmapped reads by SAMtools (-F 1804 -f 2)[57], PCR artefacts were excluded by the MarkDuplicates.jar from Picard-tools-1.119. The BamCoverage function of deepTools with normalization to RPGC was used to generate the bigwig files for visualization of ATAC-seq reads in the genome browser IGV (-bs 20−smoothLength 40−normalizeUsing RPGC−effectiveGenomeSize 2150570000 -e). Afterwards, fastq.gz files of each of the three replicates were merged and processing was

performed like described. Peak calling was applied from merged bam-files with MACS2 (−q 0.01–nomodel–shift −75–extsize150)[58], whereas peaks overlapping a blacklist defined by ENCODE were discarded with the help of Bedtools (intersect -wa)[59]. Called peaks from merged files and individual bam files were further used to quantify and normalize counts as well as to calculate differentially accessible chromatin regions upon *Lmna* deletion using the R package DiffBind with normalizing to RPKM and using the DESeq2 method to perform differential analysis. Peaks were annotated by a custom R script combining ChIPseeker[60] and rtracklayer packages[61]. Promoter regions were defined as ±3 kb around the mm10 gene transcription start site unless otherwise stated.

Principle component analysis (PCA), correlation heatmap, heatmap of 10 000 most significant peaks and annotation pie chart were generated by the help of the DiffBind package. Volcano plots are based on the EnhancedVolcano package[62]. Average ATAC-Seq tag intensities in the ATAC-seq normalized mapped reads on up- and down-regulated genes were calculated using ngs.plot.r (-G mm10 -R tss -C contig.txt -O out_TSS_ngs5kb -L 5000 -FL 200) and the output matrix was loaded in excel to perform the final plots. Homer motif enrichment analysis was performed with the findMotifsGenome.pl (-size 200 -len 8 -mask) function.

### Hi-C and data analysis
Hi-C libraries were generated according to[63] and sequenced paired end with read length of 75 on NextSeq™ 550-System (Illumina). Hi-C paired-end reads were aligned to the mm10 genome, duplicate reads were removed, reads were assigned to MboI restriction fragments, filtered into valid interactions, and the interaction matrices were generated using the HiC-Pro pipeline default settings[64]. HiC-Pro valid interaction reads were then used to detect significant loops and generate Washu epigenome browser file using FitHiChIP (FDR 0.01 Hi-C mode binsize 10 LowDistThre 20000 UppDistThr 2000000). PE alignment output of HiC-Pro was used to build a *h5* matrix of 100Kb using hicBuildMatrix command with default settings from HiCExplorer. The *h5* matrixes were normalized (hiNormalize command), corrected (HicCorrectMatrix command) and matrix bins were merged (hicMergeMatrixbins -nb 50). The merged bins matrix was then used to generate heatmaps. Valid pairs HiC-Pro output was used to generate HOMER TagDirectories[65] which were used to perform the principal component analysis with a resolution of 25 kb. DNA segments with a positive principal component 1 (PC1) value in at least two biological replicates were defined as compartment A, while those with a negative value were defined as compartment B. DNA compartment switch from A to B or B to A was considered, if changes were observed in at least two biological replicates in both control and Lamin A/C KO. A DNA segment was defined as static if it stayed in compartment A or B in at least two biological replicates from the control and Lamin A/C KO mESCs.

### Statistics and Reproducibility
All experiments were performed at least three independent times and the respective data were used for statistical analyses. Values for n represent always biologically independent samples. Data are presented as mean ± SD. Differences between groups were assessed using an unpaired two-tailed Student's t-test or ANOVA multiple comparisons, as indicated in the figure legends. For small size samples (*n* = 3) which cannot meet the normal distribution, differences between groups were assessed using the nonparametric Mann-Whitney U Test or Kruskal-Wallis Test for multiple comparisons. *P* values represent significance *$P < 0.05$, **$P < 0.01$, ***$P < 0.001$. GraphPad Prism v8.0.2 was used to perform the statistical analysis and prepare figures.

Images in Fig. 2b, d, Fig. 3c, 5d, e, Fig. 8o, Supplementary Fig. 1c, Supplementary Fig. 3a–c, Supplementary Fig. 3i, Supplementary Fig. 6a, b, Supplementary Fig. 6d, Supplementary Fig. 8a, b are representative examples of at least three independent experiments with

similar results. For Figs. 2a, 5b, c, Fig. 8p, Supplementary Fig.1b, Supplementary Fig. 2h, Supplementary Fig.8c at least three independent experiments with similar results were performed and representative WB images are shown in the figures.

For the box plots in Fig. 5a, the center line indicates the median, upper bound the 75th percentile, lower bound the 25th percentile and the whiskers extend to 1.5 IQR (interquartile range). Values of n represent all *Lmna* transcript variants detected in the sequencing datasets indicated in the figure legend. Differences between groups were assessed using one-way ANOVA with Tukey correction multiple comparisons.

### Reporting summary
Further information on research design is available in the Nature Research Reporting Summary linked to this article.

### Data availability
Raw RNA-seq, ATAC-seq and Hi-C data generated in this study have been deposited in GEO database under accession code GSE164069 . Processed RNA-seq, ATAC-seq and Hi-C data are provided in the Supplementary Information. Lamin A DamID (GSE62685)[25] and Lamin B1 DamID data (GSE17051)[3], expression data during early mouse (GSE57249) and human embryonic development (GSE36552 and GSE101571)[66–68], mESC, mESC-CMs (GSE47948), hiPSC and hiPSC-CM (GSE107654)[69] as well as patients with *LMNA*-associated DCM mutation (GSE120836)[28] were retrieved from previously published studies. Source data are provided with this paper.

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

## Acknowledgements

We thank Alexandra Buse for technical assistance. Further, we like to thank Stefanie Uhlig, the FlowCore Mannheim and the Institute of Transfusion Medicine and Immunology, for excellent support. GD was supported by the DZHK (81Z0500202), funded by BMBF, the CRC 1366 (Project A03), the CRC 1550 (Project A03), the CRC 873 (Project A16) funded by the DFG and the Baden-Württemberg foundation special program "Angioformatics single cell platform".

## Author contributions

Y.W., A.E., L.K., O.L., H.Sh., H.Se., F.T. and MMM performed the experiments. J.C., A.E. and L.K. performed the bioinformatic analysis. I.E., J.B., T.W. and J.H. provided reagents and valuable intellectual input. Y.W. and G.D. designed the experiments, analyzed the data and wrote the manuscript. All authors discussed the results and commented on the manuscript.

## Funding

## Competing interests

The authors declare no competing interests.
