## [Peer Review File · Nature Communications]

Lamin A/C-dependent chromatin architecture safeguards naïve pluripotency to prevent aberrant cardiovascular cell fate and functionREVIEWER COMMENTS

Reviewer #1 (Remarks to the Author):

This manuscript dissects the contributions of the disease-linked nuclear structural protein Lamin A to cell fate establishment and maintenance through its roles in guiding 3D genome organization. Mutations in the LMNA gene give rise to cardiomyopathy syndromes, and several other groups have focused on the effects of LMNA loss or mutation on differentiated cardiomyocytes (e.g. the work of Rajan Jain, Jonathan Epstein, Joseph Wu, and Charles Murry). Wang and colleagues significantly advance the conversation in the field by demonstrating that LMNA disruption has major effects on gene expression and cell fate choices both early in development and in differentiated cells. This work challenges the prevailing model that Lamin A is only expressed in and regulates the function of differentiated cells, and will likely catalyze future studies focused on the under-studied roles of Lamin A in early development. A second major strength of this paper is the identification of GATA4-directed transcription dysregulation in LMNA $-/-$ cells, and the demonstration that modulating GATA4 dose counterbalances the effects of LMNA loss.

Overall the data presented are very strong and bring new perspectives to the field. The critiques that follow are focused on improving clarity and flow of the manuscript.

Major Critiques:

1. HiC and ATAC-seq analyses: it is quite strange that 3D genome organization and accessibility changes are largely restricted to chromosome 14 in LMNA $-/-$ cells. Have the authors performed karyotyping to confirm that the mESCs are euploid? More generally, how do the authors reconcile their findings that LMNA $-/-$ ESCs exhibit 3D genome disorganization and transcriptional changes with previously published work showing that mESCs lacking all 3 Lamin genes (LMNA, LMNB1, LMNB2) have only very modest disruptions of genome organization and gene expression (see Zheng et al., Mol Cell 2018, PMID 30201095)? Some discussion of the similarities and contrasts here would be helpful to the field.
2. It is fascinating that LMNA disruption appears to have completely different effects depending on the stage of development where disruption occurs (Figure 7). However, I wish the authors would advance a model or speculate how this could come to be. Is this mediated by the same underlying mechanism in both contexts? Could Lamin A influence chromatin compaction / accessibility that is then acted on by distinct sets of TFs in pluripotent vs. differentiated cells?
3. Transitions between naïve and primed pluripotent states figure very importantly in Figure 7. The authors should make very clear in the figure legend and the results text exactly what culture conditions correspond to naïve vs. primed, as this is an area with some ambiguity / variability in the field.
4. The authors show on the one hand that LMNA ablation causes gene expression changes. On the other, they show that differentiated LMNA $-/-$ cardiomyocytes exhibit increased sensitivity to strain and other stressors (Figure 3). The authors do not explicitly discuss this, but taken together these findings imply that Lamin A is required both for gene regulation and for physical protection of the nucleus. Some discussion of these roles would be helpful.

Minor Critiques:

1. The organization of the results section is a little odd. Instead of moving back and forth between early development and mouse cardiac phenotypes, could the results section instead focus on early development and regulation of gene expression (Figs 1,4,5, and 7) before cardiac phenotypes (Figs 2,

3, and 6)?

2. There appears to be a typo in the legend and labeling of Figure 6b - the 'scramble' and 'GATA4 RNAi' conditions are incorrectly labeled.

Reviewer #2 (Remarks to the Author):

The manuscript by Wang et al examines the effects of Lamin A deficiency in cardiac lineage development in embryonic stem cells (ESCs) and postnatal mouse hearts. Extensive analyses of chromatin compaction and cardiac gene expression support a mechanism whereby loss of Lamin A leads to precocious activation of cardiac genes and premature differentiation of ESC-derived cardiomyocytes. Supporting *in vivo* studies of lamin A^{-/-} mice show that post-natal day 1 (P1) cardiomyocytes have reduced size and proliferation with increased cardiac contractility and decreased capillary density. Together these studies support critical roles for lamin A in ESC cardiac lineage gene expression as well as postnatal cardiomyocyte maturation.

Overall the data are clear and convincing, the manuscript is well written and novel insights into Lamin A function in cardiac progenitors and postnatal cardiomyocytes are reported. In general the conclusions are well supported by the data with the exception that one would expect cardiac developmental anomalies *in vivo* if the reported critical functions in cardiac differentiation and gene expression are occurring as seen in the context of cultured ESC. It is not entirely clear that the dilated cardiomyopathy arises directly from the embryonic lineage abnormalities from the data presented. It seems possible Lamin A has critical functions in perinatal cardiomyocytes that contribute to maturation defects and dilated cardiomyopathy at later stages.

Major Comment.

1. As mentioned above, my main criticism of the study is the lack of information on prenatal heart development in the Lamin a^{-/-} mice. Is there any evidence of prenatal lethality or abnormalities in heart formation as would be expected with major shifts in cardiogenic gene expression? It seems important to examine E7.5-9.5 embryos to determine if the timing of embryonic cardiac differentiation or heart formation is affected by loss of Lamin A *in vivo*.

Minor Comments

2. It would be nice to see the cell morphologies and maturation of sarcomeres in images of individual cardiomyocytes and endothelial cells with immunofluorescence in the ESC experiments.

3. Individual data points should be included in the histograms. Also, Student's t-tests are not appropriate with small sample sizes used. For some mouse studies, n=3 sample sizes were used which is minimal.

4. It would be helpful to have GEO accession numbers for the data sets analyzed in the study as described in the data and materials availability section on page 30.

5. CPC is used as an abbreviation for cardiac progenitor cells in the ESC system. It should be defined at first use as the stage when these progenitors arise, but the cardiac progenitors were not actually sorted or purified. The authors might consider using another term since CPC was used extensively in discredited literature to describe c-kit⁺ adult cardiac cells.

6. The current work seems to contradict a previous report (2021, JCI: reference #19) that the H222P mutant form of Lamin A caused prenatal cardiac abnormalities in mice and delayed differentiation of hESC. How do these published data relate to the current study?

7. In Figure 7, RNAseq data from hESC-CMs and human LMNA-DCM cardiac tissue are compared. The human tissue is from whole hearts, thus the cell types being analyzed are not completely comparable. The conclusion paragraph on p14 is a bit overstated in that the connections between the lamin A functions in embryonic stem cells have not been directly linked to the cardiomyocyte abnormalities in the postnatal mouse or adult human hearts.

Reviewer #3 (Remarks to the Author):

In this manuscript, Wang and colleagues describe the role of lamin A in maintaining chromatin organization in embryonic stem cells (ESCs). The authors report that chromatin dysregulations in *Lmna*^{-/-} ESCs lead to activation of transcriptional signature-related cardiomyocytes. Moreover, a contrary function of lamin A is observed in naïve pluripotent stem cells and cardiomyocytes. Overall, the authors conclude that lamin A plays a role in maintaining chromatin architecture in pluripotent stem cells.

Below are the comments on the manuscript:

-Recent studies suggest that lamin a and c have distinct roles in LAD dynamics. The only *Lamin c* expressing mice are entirely normal, whereas *Lamina*^{-/-} mice die around 4-7 weeks of age. In the current study, both lamin a and c are deleted. So, it is essential to show the transcriptomic signature differences in individual knockout cells and show which isoform is the cause for the observed phenotypes.

-Guenantin et al. reported that mESCs with *Lmna*^{H222p/+} mutation had delayed cardiogenesis program and low embryo body beating activity. However, the results from the current study demonstrate an opposite phenotype. The authors need to clarify this discrepancy.

-It is not clear from the methods which *Lmna* mutant mouse model is used for the in vivo study.

-Chromatin organization is also disrupted in progeroid syndromes such as Werner syndrome. Therefore, it is important to check whether dysregulation of cardiac genes is also found in Werner syndrome or specific to only laminopathies.

In figure 7a, human blastocysts were used to check the lamin a level during embryogenesis, which included both ICM and trophectoderm cells. The authors should use only ICM or ESCs similar to the mouse.

-In many places, the authors mention "Lamin A loss" and are referring to protein, so the current way it is written is wrong.

The main conclusion of the study is about the role of the lamina in pluripotent stem cells. There is a disconnection between results from the lamina mutant model and stem cells. The results from the current study may be suitable for a specific journal.

Point-by-point response to the reviewers' comments

We would like to express our sincere gratitude to all the reviewers for their appreciation of our work and especially for their thoughtful and constructive comments, which helped us to improve the quality of our manuscript considerably and to clarify a number of important points. To address the reviewers' concerns we have performed a number of additional experiments, as detailed in the following point-by-point response.

In addition, we have restructured the manuscript to improve the flow as suggested by one of the reviewers. We now present the studies addressing the role of lamin A/C in cardiac differentiation using mESCs and hiPSCs model systems in the first five figures, while the data on the phenotype of *Lmna*^{+/-} and *Lmna*^{-/-} mice during embryogenesis and postnatal life are presented in Fig 6 (new figure studying the role of lamin A/C in heart development) and Fig. 7-9.

Figure in the initial manuscript	Figure in the revised manuscript
Figure 1	Figure 1
Figure 2	Figure 7
Figure 3	Figure 8
Figure 4	Figure 2
Figure 5	data split between Figure 3 and Figure 4
Figure 6	Figure 9
Figure 7	Figure 5
Figure 8	Figure 10

The reviewer's comments are in italics.

Reviewer #1:

Reviewer #1 (Remarks to the Author):

This manuscript dissects the contributions of the disease-linked nuclear structural protein Lamin A to cell fate establishment and maintenance through its roles in guiding 3D genome organization. Mutations in the LMNA gene give rise to cardiomyopathy syndromes, and several other groups have focused on the effects of LMNA loss or mutation on differentiated cardiomyocytes (e.g. the work of Rajan Jain, Jonathan Epstein, Joseph Wu, and Charles Murry). Wang and colleagues significantly advance the conversation in the field by demonstrating that LMNA disruption has major effects on gene expression and cell fate choices both early in development and in differentiated cells. This work challenges the prevailing model that Lamin A is only expressed in and regulates the function of differentiated cells, and will likely catalyze future studies focused on the under-studied roles of Lamin A in early development. A second major strength of this paper is the identification of GATA4-directed transcription dysregulation in LMNA $-/-$ cells, and the demonstration that modulating GATA4 dose counterbalances the effects of LMNA loss.

Overall the data presented are very strong and bring new perspectives to the field. The critiques that follow are focused on improving clarity and flow of the manuscript.

Response: We appreciate that the reviewer finds our study strong and important for the field. In the revised version of the manuscript, we have further extended and strengthened the analysis of the impact of lamin A/C in cardiac development and have improved the flow of the manuscript as suggested by the reviewer.

Major Critiques:

1. HiC and ATAC-seq analyses: it is quite strange that 3D genome organization and accessibility changes are largely restricted to chromosome 14 in LMNA $-/-$ cells. Have the authors performed karyotyping to confirm that the mESCs are euploid? More generally, how do the authors reconcile their findings that LMNA $-/-$ ESCs exhibit 3D genome disorganization and transcriptional changes with previously published work showing that mESCs lacking all 3 Lamin genes (LMNA, LMNB1, LMNB2) have only very modest disruptions of genome organization and gene expression (see Zheng et al., Mol Cell 2018, PMID 30201095)? Some discussion of the similarities and contrasts here would be helpful to the field.

Response: In the revised manuscript, we have now included whole-chromosome painting with a probe for Chr. 14 of interphase and metaphase cell preparations, showing large-scale chromatin decondensation of Chr. 14 in the interphase nucleus and expanded chromosome territory in *Lmna*^{-/-} ESCs compared to control cells, while the chromosome number was not changed (Supplementary Fig.1c). In addition, we have now included genome tracks of ATAC-Seq and RNA-Seq reads showing large-scale chromatin opening, which co-occurs with dramatic upregulation of transcriptional activity within a large portion of chromosome 14 (but not the whole chromosome, Fig. 3f). The represented tracks are merged ATAC-Seq experiments of three control and three *Lmna* knockout clones generated using the CRISPR/Cas9 gene editing of the same parental line. Similar decompaction of Chr. 14 was observed in Zheng *et al.*, *Mol Cell* 2018, PMID 30201095.

In our revised manuscript, we have discussed these results on page 20, lines 497-503, as follows:

Together with specific changes in chromatin organization, we observed large-scale changes in chromatin compaction and 3D organization of chromosome 14. In line with these findings, major changes were observed in the compaction of chromosome 1, 4, 13 and 14 in lamin B1, B2, and A/C triple KO mouse mESCs¹, suggesting that lamin A/C might play a specific role in chromosome 14 compaction, whereas lamin B1 and B2 could be involved in compaction of chromosomes 1, 4 and 13.

In addition, similar to the Zheng *et al* study we did not notice major changes in Hi-C contact matrices on other chromosomes, but we did observe chromatin compartments switches, which were significantly enriched for genes involved in calcium ion transmembrane transport, chromatin organization and muscle cell differentiation.

Moreover, we did not observe chromatin decompaction in ESCs harboring the *Lmna* p.H222P mutation, causing Emery-Dreifuss muscular dystrophy and cardiomyopathy, and the *Lmna* p.G609G mutation, resulting in Hutchinson-Gilford progeria syndrome generated using the CRISPR/Cas9 gene editing of the same parental line (Supplementary Fig. 4j)

2. It is fascinating that LMNA disruption appears to have completely different effects depending on the stage of development where disruption occurs (Figure 7). However, I wish the authors would advance a model or speculate how this could come to be. Is this mediated by the same underlying mechanism in both contexts? Could Lamin A influence chromatin compaction / accessibility that is then acted on by distinct sets of TFs in pluripotent vs. differentiated cells?

We thank the reviewer for this comment. In the revised manuscript, we have performed a series of experiments addressing this interesting aspect. In brief, we analyzed chromatin accessibility in wild-type and *Lmna*^{-/-} ESCs using ATAC-Seq and observed a widespread increase in chromatin accessibility across the genome already in ESCs (Fig. 3f-h). Cluster analysis revealed that a large set of genes that showed increased chromatin accessibility in ESCs were either upregulated in all stages of cardiac differentiation (Fig. 3i, cluster A) or specifically in ESCs, CPs or CMs (Fig. 3i, clusters B-G). Intersection of the ATAC-Seq and RNA-Seq analysis revealed that more than half of the genes upregulated upon lamin A/C loss in ESCs showed increased chromatin accessibility (Fig. 3j) and GO analysis of the overlapping genes indicated overrepresentation of GO terms linked to transcription, cell differentiation, outflow tract morphogenesis as well as Wnt signaling (Fig. 3k). Motif enrichment analysis within chromatin regions of genes upregulated upon lamin A/C loss of function and characterized with more open chromatin identified binding motifs for Klf, Sox and Tead family members as well as Gata4 binding motifs (Fig. 3l). Interestingly, intersection of ATAC-Seq in ESCs with the RNA-Seq analysis from CPs and CMs, revealed a very high overlap of genes that undergo chromatin opening as a result of lamin A/C loss in ESCs and genes that are upregulated only later during differentiation, i.e. in *Lmna*^{-/-} CPs and CMs (Fig. 3m). GO analysis of genes showing open chromatin conformation already at the ESC stage and upregulation in CPs and CMs revealed overrepresentation of GO terms linked to heart morphogenesis and cell fate commitment (Fig. 3n), supporting the notion that lamin A/C deficiency in ESCs primes cardiac specific genes for expression in later stages during development. Motif enrichment analysis within chromatin regions of genes upregulated upon lamin A/C loss of function in CPs and CMs showing increase in chromatin accessibility (epigenetic priming) already in ESCs, identified binding motifs for Tead and Gata family members as well as Meis1, which regulates postnatal cardiomyocyte cell cycle arrest and Foxo1, shown to be activated and to contribute to the pathogenesis of cardiac laminopathies (Fig. 3o).

We next compared the role of lamin A/C for chromatin organization in CMs to that in ESCs. Importantly, we found 55% overlap between genes that showed increased chromatin accessibility in both *Lmna*^{-/-} CMs and ESCs, whereas 45% showed more open chromatin only in CMs (Fig. 4f, Supplementary Fig. 5g). In summary, the analysis presented in Fig. 4f-h indicate that lamin A/C plays a key role in transcriptional priming of genes involved in cardiac development, CM contraction and Ca²⁺ homeostasis in ESCs, whereas in CMs lamin A/C specifically regulates genes involved in cardiac contraction and sarcomere organization.

Further, we present data suggesting that the different effects depending on the stage of cardiogenesis where disruption occurs is dependent on the important function of lamin A/C in tethering and silencing CM specific genes at the nuclear periphery and thereby preventing

epigenetic priming of ESCs resulting in precocious cardiomyocyte differentiation. In CMs in contrast to ESCs, CM-specific genes are already located in the transcriptionally permissive nuclear interior ² (Fig. 6c), while in ESCs and non-CMs they are embedded in the non-permissive nuclear periphery similar to ESCs. This notion is further supported by the fact that *Lmna* mutations, which do not affect the tethering of CM-genes at the nuclear periphery in ESCs, show impaired CM differentiation similar to the specific depletion of lamin A/C in cardiomyocytes.

3. Transitions between naïve and primed pluripotent states figure very importantly in Figure 7. The authors should make very clear in the figure legend and the results text exactly what culture conditions correspond to naïve vs. primed, as this is an area with some ambiguity / variability in the field.

In the figure legend of Figure 5 (old Figure 7) we have now included a reference to Supplementary Fig. 6a, which shows a schematic representation of the strategy used for the conversion of primed human iPSCs into naïve hiPSC. In addition, we have now provided more detailed description of naïve vs. primed culture conditions in the methods section: page 27, lines 692-704 and page 28, lines 705-706.

4. The authors show on the one hand that LMNA ablation causes gene expression changes. On the other, they show that differentiated LMNA^{-/-} cardiomyocytes exhibit increased sensitivity to strain and other stressors (Figure 3). The authors do not explicitly discuss this, but taken together these findings imply that Lamin A is required both for gene regulation and for physical protection of the nucleus. Some discussion of these roles would be helpful.

In our revised manuscript, we have discussed these results on page 22, lines 563-576:

In addition, the premature cell cycle arrest and elevated cell death could be due to the inability of *Lmna*^{-/-} CMs to respond adequately to oxidative and mechanical stress. Indeed, mechanosensing by the nuclear lamina have been shown to protect against nuclear rupture, DNA damage, and cell-cycle withdrawal, while the oxygen-rich postnatal environment was proposed to induce cell-cycle arrest through the DNA damage response pathway ^{31,49}. Interestingly, we observed increased DNA damage and cell death in *Lmna*^{-/-} non-CMs subjected to stretch and oxidative stress, suggesting that activation of the DNA damage pathway might contribute to the premature cell cycle arrest and increased cell death upon lamin A/C LOF. Importantly, during embryogenesis the RV and the LV eject blood at a high pressure into the systemic circulation as blood shunts through the ductus arteriosus and foramen ovale, while after birth the pulmonary circulation is a low-pressure circuit. Thus, the

differences observed in RV and LV function during embryogenesis and after birth might be due to the important function of lamin A/C in mechanosensing and response.

Minor Critiques:

1. The organization of the results section is a little odd. Instead of moving back and forth between early development and mouse cardiac phenotypes, could the results section instead focus on early development and regulation of gene expression (Figs 1,4,5, and 7) before cardiac phenotypes (Figs 2, 3, and 6)?

We agree with the reviewer. We have restructured the manuscript to improve the flow. We now present the studies addressing the role of lamin A/C in cardiac differentiation using mESCs and hiPSCs model systems in the first five figures, while the data on the phenotype of *Lmna*^{+/-} and *Lmna*^{-/-} mice during embryogenesis and postnatal life are presented in Fig 6 (new figure studying the role of lamin A/C in heart development) and Fig. 7-8.

Figure in the initial manuscript	Figure in the revised manuscript
Figure 1	Figure 1
Figure 2	Figure 7
Figure 3	Figure 8
Figure 4	Figure 2
Figure 5	data split between Figure 3 and Figure 4
Figure 6	Figure 9
Figure 7	Figure 5
Figure 8	Figure 10

2. There appears to be a typo in the legend and labeling of Figure 6b - the 'scramble' and 'GATA4 RNAi' conditions are incorrectly labeled.

We thank the reviewer for pointing out this mistake. We have now properly labeled the different conditions in Fig. 9b (old Fig. 6b).

Reviewer #2 (Remarks to the Author):

The manuscript by Wang et al examines the effects of Lamin A deficiency in cardiac lineage development in embryonic stem cells (ESCs) and postnatal mouse hearts. Extensive analyses of chromatin compaction and cardiac gene expression support a mechanism whereby loss of Lamin A leads to precocious activation of cardiac genes and premature differentiation of ESC-derived cardiomyocytes. Supporting in vivo studies of lamin A^{-/-} mice show that post-natal day 1 (P1) cardiomyocytes have reduced size and proliferation with increased cardiac contractility and decreased capillary density. Together these studies support critical roles for lamin A in ESC cardiac lineage gene expression as well as postnatal cardiomyocyte maturation.

Overall the data are clear and convincing, the manuscript is well written and novel insights into Lamin A function in cardiac progenitors and postnatal cardiomyocytes are reported. In general the conclusions are well supported by the data with the exception that one would expect cardiac developmental anomalies in vivo if the reported critical functions in cardiac differentiation and gene expression are occurring as seen in the context of cultured ESC. It is not entirely clear that the dilated cardiomyopathy arises directly from the embryonic lineage abnormalities from the data presented. It seems possible Lamin A has critical functions in perinatal cardiomyocytes that contribute to maturation defects and dilated cardiomyopathy at later stages.

Response: We thank the reviewer for his/her appreciation of our study. In the revised version of the manuscript, we have further extended our analysis to address the impact of lamin A/C in cardiac development as suggested by the reviewer.

Major Comment.

1. As mentioned above, my main criticism of the study is the lack of information on prenatal heart development in the Lamin a^{-/-} mice. Is there any evidence of prenatal lethality or abnormalities in heart formation as would be expected with major shifts in cardiogenic gene expression? It seems important to examine E7.5-9.5 embryos to determine if the timing of embryonic cardiac differentiation or heart formation is affected by loss of Lamin A in vivo.

Response: We thank the reviewer for this comment. In the revised manuscript we have included a new figure (Fig. 6) addressing the importance of lamin A/C during heart development (page 14, lines 363-370 and page 15, lines 371-386, and 389-392). In brief, we

now present expression analysis showing significant upregulation of cardiac progenitor (CP) and cardiomyocyte (CM) marker genes in dissected pharyngeal mesoderm and hearts of E8.5 embryos as well as E9.5 hearts upon *Lmna* ablation, consistent with our in vitro cell culture based studies (Fig. 6a, 6b). In addition, we present FISH analysis showing that CMs specific genes are located in the nuclear interior of *Lmna*^{+/+} CMs and at the nuclear periphery in *Lmna*^{+/+} fibroblasts respectively, while they are found in the nuclear interior in both CMs and fibroblasts isolated from *Lmna*^{+/-} and *Lmna*^{-/-} embryos, supporting a role of lamin A/C in tethering CM specific genes to the nuclear periphery in non-CMs (Fig. 6c, 6d). Functional analysis revealed significantly increased right ventricular ejection fraction, an index of cardiac contractility in *Lmna*^{-/-} embryos (Fig. 6e, 6f), while LV function was not significantly affected. Histological examination unveiled non-compaction of the RV myocardium in both *Lmna*^{+/-} and *Lmna*^{-/-} embryos, while only *Lmna*^{-/-} embryos showed LV non-compaction (Fig. 6g, 6h). Moreover, we found increased binucleation and decreased proliferation of CMs as well as significantly decreased capillary density in hearts from both *Lmna*^{+/-} and *Lmna*^{-/-} E18.5 embryos (Fig. 6i-p).

We have also included discussion on page 21, lines 538-551 addressing the more pronounced RV phenotype in *Lmna*^{+/-} and *Lmna*^{-/-} embryos, which may be accounted by the different embryological origin of the RV compared to the LV myocardium, the different contribution of the second heart field progenitor cells, which give rise to the RV, to the distinct cell types in the heart and the possible involvement of *Isl1*, a pioneer TF in second heart field^{5,6}, which is also found in lamin A/C LADs in ESCs. We have also included discussion regarding the differences observed in RV and LV function during embryogenesis and after birth, which might be connected to the function of lamin A/C in mechanosensing and response – on page 22, lines 563-576.

Minor Comments

2. It would be nice to see the cell morphologies and maturation of sarcomeres in images of individual cardiomyocytes and endothelial cells with immunofluorescence in the ESC experiments.

Response: We have now included immunostainings of ESC- derived cardiomyocytes and endothelial cells in Supplementary Fig. 8a, b.

3. Individual data points should be included in the histograms. Also, Student's t-tests are not appropriate with small sample sizes used. For some mouse studies, n=3 sample sizes were used which is minimal.

Response: In our revised manuscript, we have included data points in all histograms (in most functional animal studies n=6 have been now used). The statistical analysis used in each panel and the justification of the utilized statistical method have been thoroughly outlined in the Statistical Analysis paragraph in the methods section (page 37, lines 961-971).

4. It would be helpful to have GEO accession numbers for the data sets analyzed in the study as described in the data and materials availability section on page 30.

Response: The GEO accession number GSE164069 indicated in the Data and materials availability section includes raw and processed data of all NGS experiments: RNA-Seq, ATAC-Seq and HiC-Seq of ESCs, CP, CMs and embryoid bodies. We have created a reviewers token, to enable the reviewer to assess data completeness.

To review GEO accession GSE164069:

Please, go to <https://www.ncbi.nlm.nih.gov/geo/query/acc.cgi?acc=GSE164069>

Enter token kjqdwggbjqzxeb into the box.

We have now included GEO accession numbers for all previously published studies, used in our analysis in the Data and materials availability section on page 37, lines 955-960.

5. CPC is used as an abbreviation for cardiac progenitor cells in the ESC system. It should be defined at first use as the stage when these progenitors arise, but the cardiac progenitors were not actually sorted or purified. The authors might consider using another term since CPC was used extensively in discredited literature to describe c-kit+ adult cardiac cells.

Response: We were unaware that this abbreviation would lead to a wrong association. We have now used the abbreviation CP for cardiac progenitor throughout the text.

6. The current work seems to contradict a previous report (2021, JCI: reference #19) that the H222P mutant form of Lamin A caused prenatal cardiac abnormalities in mice and delayed differentiation of hESC. How do these published data relate to the current study?

Response: We thank the reviewer for this comment, which prompted us to further characterize the mechanisms behind these largely different phenotypes. In our revised manuscript we have included characterization of ESC lines harboring the Lmna p.H222P mutation, causing Emery-Dreifuss muscular dystrophy and cardiomyopathy, and the Lmna p.G609G mutation, resulting in Hutchinson-Gilford progeria syndrome (accelerated aging and premature death due to cardiovascular events) (Supplementary Fig. 2a-f, Supplementary Fig.

3i, j, Supplementary Fig. 4i-k). ESCs harbouring these two distinct mutations showed very different differentiation behavior. We observed significantly decreased expression of cardiac mesoderm marker genes, such as *Eomes* and *Mesp1*, at mesoderm stage (d4) (Supplementary Fig. 2c), cardiac progenitor (CP) markers at CP stage (Supplementary Fig. 2d) and CM genes at day 8 (Supplementary Fig. 2e). Consistent with the expression data we observed significantly decreased number of CMs (Supplementary Fig. 2f). These data are in line with the previously published study, which showed that ESC lines harboring a *Lmna* p.H222P mutation have impaired cardiac differentiation, and suggest that the molecular mechanisms resulting in heart disease as a result of distinct point mutations in the *Lmna* gene are different. To further characterize the mechanisms behind these distinct phenotypes, we performed FISH and ATAC-Seq analysis of *Lmna*^{H222P/H222P}, *Lmna*^{G609G/+} and *Lmna*^{G609G/G609G} ESCs. FISH analysis revealed that in contrast to *Lmna* knockout ESCs, cardiomyocyte-specific genes were still localized at the nuclear periphery in ESC lines carrying *Lmna* p.H222P and *Lmna* p.G609G mutations, suggesting that dissociation of CM genes from the nuclear periphery might be important for the precocious CM differentiation observed upon lamin A/C loss of function (Supplementary Fig. 3i, j). Further, in contrast to *Lmna* knockout ESCs, *Lmna*^{H222P/H222P}, *Lmna*^{G609G/+} and *Lmna*^{G609G/G609G} ESCs did not show obvious increase in genome-wide chromatin accessibility or at Chr. 14 as well as epigenetic priming at cardiac-specific loci, supporting further the notion that the molecular mechanisms underlying the cellular phenotypes upon lamin A/C loss of function and mutation are different (Supplementary Fig. 4i-k).

7. In Figure 7, RNAseq data from hESC-CMs and human LMNA-DCM cardiac tissue are compared. The human tissue is from whole hearts, thus the cell types being analyzed are not completely comparable. The conclusion paragraph on p14 is a bit overstated in that the connections between the lamin A functions in embryonic stem cells have not been directly linked to the cardiomyocyte abnormalities in the postnatal mouse or adult human hearts.

Response: We agree with the reviewer that the RNA-seq data from ESC-derived CMs and human *LMNA*-DCM cardiac tissue can not be directly compared (initial Figure 7, in the revised manuscript Figure 5). With this experiment, we wanted to make a point that the expression changes in cardiac structural and contraction genes observed in CMs derived from *Lmna*^{-/-} ESCs, but not by specific deletion in already committed CMs, correlate with the expression changes in patients with *LMNA*-associated DCM, characterized by significantly lower lamin A/C protein levels. The new data presented in Fig. 6 now corroborate our in vitro cell-culture based findings as outlined in our response to the major comment of the reviewer.

Reviewer #3 (Remarks to the Author):

*In this manuscript, Wang and colleagues describe the role of lamin A in maintaining chromatin organization in embryonic stem cells (ESCs). The authors report that chromatin dysregulations in *Lmna*^{-/-} ESCs lead to activation of transcriptional signature-related cardiomyocytes. Moreover, a contrary function of lamin A is observed in naïve pluripotent stem cells and cardiomyocytes. Overall, the authors conclude that lamin A plays a role in maintaining chromatin architecture in pluripotent stem cells.*

Below are the comments on the manuscript:

-Recent studies suggest that lamin a and c have distinct roles in LAD dynamics. The only Lamin c expressing mice are entirely normal, whereas Lamina^{-/-} mice die around 4-7 weeks of age. In the current study, both lamin a and c are deleted. So, it is essential to show the transcriptomic signature differences in individual knockout cells and show which isoform is the cause for the observed phenotypes.

Response: We thank the reviewer for this comment. In the revised manuscript, we have included data suggesting a function of lamin A rather than lamin C in tethering cardiac-specific genes to the nuclear periphery as well as repressing genes upregulated in *Lmna*^{-/-} ESCs. These data are described on page 11, lines 263-266 and presented in Supplementary Fig. 4I-o.

However, since *LMNA* mutations resulting in haploinsufficiency affect both lamin A and C and the mouse model that we have used in our manuscript results in haploinsufficiency/deficiency of both proteins we believe that our study brings novel important insights in the mechanisms at the roots of cardiomyopathies due to *LMNA* loss-of-function.

*-Guenantin et al. reported that mESCs with *Lmna*^{H222p/+} mutation had delayed cardiogenesis program and low embryo body beating activity. However, the results from the current study demonstrate an opposite phenotype. The authors need to clarify this discrepancy.*

Response: We thank the reviewer for this comment, which was also raised by another reviewer of our manuscript. The indicated discrepancy, prompted us to further characterize the mechanisms behind these largely different phenotypes. In our revised manuscript we have included characterization of ESC lines harboring the *Lmna* p.H222P mutation, causing Emery-Dreifuss muscular dystrophy and cardiomyopathy, and the *Lmna* p.G609G mutation,

resulting in Hutchinson-Gilford progeria syndrome (accelerated aging and premature death due to cardiovascular events) (Supplementary Fig. 2a-f, Supplementary Fig. 3i, j, Supplementary Fig. 4i-k). ESCs harbouring these two distinct mutations showed very different differentiation behavior. We observed significantly decreased expression of cardiac mesoderm marker genes, such as *Eomes* and *Mesp1*, at mesoderm stage (d4) (Supplementary Fig. 2c), cardiac progenitor (CP) markers at CP stage (Supplementary Fig. 2d) and CM genes at day 8 (Supplementary Fig. 2e). Consistent with the expression data we observed significantly decreased number of CMs (Supplementary Fig. 2f). These data are in line with the previously published study, which showed that ESC lines harboring a *Lmna* p.H222P mutation have impaired cardiac differentiation, and suggest that the molecular mechanisms resulting in heart disease as a result of distinct point mutations in the *Lmna* gene are different. To further characterize the mechanisms behind these distinct phenotypes, we performed FISH and ATAC-Seq analysis of *Lmna*^{H222P/H222P}, *Lmna*^{G609G/+} and *Lmna*^{G609G/G609G} ESCs. FISH analysis revealed that in contrast to *Lmna* knockout ESCs, cardiomyocyte-specific genes were still localized at the nuclear periphery in ESC lines carrying *Lmna* p.H222P and *Lmna* p.G609G mutations, suggesting that dissociation of CM genes from the nuclear periphery might be important for the precocious CM differentiation observed upon lamin A/C loss of function (Supplementary Fig. 3i, j). Further, in contrast to *Lmna* knockout ESCs, *Lmna*^{H222P/H222P}, *Lmna*^{G609G/+} and *Lmna*^{G609G/G609G} ESCs did not show obvious increase in genome-wide chromatin accessibility or at Chr. 14 as well as epigenetic priming at cardiac-specific loci, supporting further the notion that the molecular mechanisms underlying the cellular phenotypes upon lamin A/C loss of function and mutation are different (Supplementary Fig. 4i-k).

-It is not clear from the methods which Lmna mutant mouse model is used for the in vivo study.

As indicated in the section Mouse lines, the *Lmna* tm1.1Yxz/J line and the *Gata4* tm1.1Sad were obtained from Jackson Laboratory. *Lmna* tm1.1Yxz/J line was generated in Yixian Zheng's lab at the Carnegie Institute (Kim Y, et al., *Biochem Biophys Res Commun* 440(1):8-13, 2013, PubMed:23998933). We have now included a reference to the publication within the material and methods section.

-Chromatin organization is also disrupted in progeroid syndromes such as Werner syndrome. Therefore, it is important to check whether dysregulation of cardiac genes is also found in Werner syndrome or specific to only laminopathies.

Response: We thank the reviewer for this comment. As discussed above, we have now included characterization of ESC lines harbouring the *Lmna* p.G609G mutation (*Lmna*^{G609G/+} and *Lmna*^{G609G/G609G}), resulting in Hutchinson-Gilford progeria syndrome (accelerated aging and premature death due to cardiovascular events) (Supplementary Fig. 2a-e, Supplementary Fig. 3i, j, Supplementary Fig. 4 i-k), showing that the molecular mechanisms underlying the cellular phenotypes upon lamin A/C loss of function and mutation are different.

In figure 7a, human blastocysts were used to check the lamin a level during embryogenesis, which included both ICM and trophectoderm cells. The authors should use only ICM or ESCs similar to the mouse.

Response: We agree with the reviewer. We have now included data on the expression of *LMNA* in the inner cell mass during human development. We have also increased the number of hiPSCs used in the expression analysis, showing a very high variation of *LMNA* expression in hiPSCs, which might be due to the different culture states as also pointed out in Figure 5 of the revised manuscript (Figure 7 in the initial version of the manuscript).

-In many places, the authors mention “Lamin A loss” and are referring to protein, so the current way it is written is wrong.

Response: We are sorry about the inaccurate reference. Indeed, as indicated in the introduction the *Lmna* gene encodes two major isoforms: lamin A and C. We have now replaced lamin A with lamin A/C when referring to the protein throughout the text.

The main conclusion of the study is about the role of the lamina in pluripotent stem cells. There is a disconnection between results from the lamina mutant model and stem cells. The results from the current study may be suitable for a specific journal.

Response: We are grateful to the reviewer and all other reviewers for their thoughtful and constructive comments, which helped us to improve the impact of our manuscript considerably. In its revised form, our manuscript conveys the following important messages:

1. Lamin A/C plays a key role in chromatin organization in embryonic stem cells, which safeguards naïve pluripotency and ensures proper cell fate choices during cardiogenesis.
2. *Lmna* haploinsufficiency and deficiency results in premature cardiomyocyte differentiation, binucleation and cell cycle arrest during development resulting in non-compaction cardiomyopathy (new Figure 6).

3. We show that the molecular mechanisms underlying the cellular phenotypes upon lamin A/C loss-of-function mutations or mutations that change protein function are different (new data in Supplementary Fig. 2a-e, Supplementary Fig. 3i, j, Supplementary Fig. 4 i-k).
4. We show that the different effects of lamin A/C loss of function depending on the stage of development where *LMNA* disruption occurs are due to the important function of lamin A/C in tethering and silencing CM specific genes at the nuclear periphery in ESCs but not in CMs (new data in Fig. 3f-o, Fig. 4 and Fig. 6c,d).
5. We show that *Gata4* is activated by lamin A/C loss and *Gata4* silencing or haploinsufficiency rescues the aberrant cardiovascular cell fate choices induced by lamin A/C deficiency.

References:

- 1 Zheng, X. *et al.* Lamins Organize the Global Three-Dimensional Genome from the Nuclear Periphery. *Molecular Cell*, doi:10.1016/j.molcel.2018.05.017.
- 2 Poleshko, A. *et al.* Genome-Nuclear Lamina Interactions Regulate Cardiac Stem Cell Lineage Restriction. *Cell* **171**, 573-587 e514, doi:10.1016/j.cell.2017.09.018 (2017).
- 3 Cho, S. *et al.* Mechanosensing by the Lamina Protects against Nuclear Rupture, DNA Damage, and Cell-Cycle Arrest. *Dev Cell* **49**, 920-935 e925, doi:10.1016/j.devcel.2019.04.020 (2019).
- 4 Puente, B. N. *et al.* The oxygen-rich postnatal environment induces cardiomyocyte cell-cycle arrest through DNA damage response. *Cell* **157**, 565-579, doi:10.1016/j.cell.2014.03.032 (2014).
- 5 Cai, C. L. *et al.* Isl1 identifies a cardiac progenitor population that proliferates prior to differentiation and contributes a majority of cells to the heart. *Dev Cell* **5**, 877-889 (2003).
- 6 Gao, R. *et al.* Pioneering function of Isl1 in the epigenetic control of cardiomyocyte cell fate. *Cell Res* **29**, 486-501, doi:10.1038/s41422-019-0168-1 (2019).

REVIEWERS' COMMENTS

Reviewer #1 (Remarks to the Author):

I would like to thank the authors for addressing each of my critiques thoughtfully and for performing additional experiments that reveal exciting new results. The comparison of LMNA-null to LMNA point mutations is a major new addition to the paper that sheds some light on discrepancy between previous studies that evaluated effects of LMNA point mutations in ESCs or CMs vs. the findings of this group in the LMNA null state. The ATACseq data are very striking and strengthen and enrich key conclusions of the paper. This paper is an impressive synthesis of work that raises important new questions for the study of laminopathies and for dissecting the roles of the lamina on chromatin organization. I wholeheartedly support its publication in this revised form.

Reviewer #2 (Remarks to the Author):

The revised manuscript has addressed my comments and is improved. Significant new data have been added, notably on heart development in the laminA^{-/-} mice, that enhance the study. Overall, the work is thorough and convincing with new and important findings related to Lamin A regulatory functions in cardiomyocytes.

Minor comment:

The second section of the results seems to be a single paragraph of more than 3 pages (lines 153-266). This should be broken up into smaller paragraphs if possible. Some of the other paragraphs also are large.

Point-by-point response to the reviewers' comments

We would like to express our gratitude to the reviewers and the editor for their appreciation of our work, as well as for the time they have invested in carefully reviewing our manuscript.

The reviewer's comments are in italics.

Reviewer #1

Reviewer #1 (Remarks to the Author):

I would like to thank the authors for addressing each of my critiques thoughtfully and for performing additional experiments that reveal exciting new results. The comparison of LMNA-null to LMNA point mutations is a major new addition to the paper that sheds some light on discrepancy between previous studies that evaluated effects of LMNA point mutations in ESCs or CMs vs. the findings of this group in the LMNA null state. The ATACseq data are very striking and strengthen and enrich key conclusions of the paper. This paper is an impressive synthesis of work that raises important new questions for the study of laminopathies and for dissecting the roles of the lamina on chromatin organization. I wholeheartedly support its publication in this revised form.

Response: We thank the reviewer for his/ hers commendation of our manuscript.

Reviewer #2

The revised manuscript has addressed my comments and is improved. Significant new data have been added, notably on heart development in the laminA-/- mice, that enhance the study. Overall, the work is thorough and convincing with new and important findings related to Lamin A regulatory functions in cardiomyocytes.

Response: We thank the reviewer for his/ hers appreciation of the revisions that we made.

Minor comment:

The second section of the results seems to be a single paragraph of more than 3 pages (lines 153-266). This should be broken up into smaller paragraphs if possible. Some of the other paragraphs also are large.

Response: We have introduced additional subheadings to break up longer sections into smaller more focused pieces:

Line 155: Lamin A/C loss induces 3D chromatin reorganization in mESCs

Line 208: Lamin A/C deficiency in ESCs primes cardiac specific gene expression

Line 269: Cell-type specific role of Lamin A/C in shaping chromatin accessibility

Line 310: Lamin A/C is crucial for naïve pluripotency

Line 326: Distinct roles of lamin A/C in naïve PSC and CMs for cardiac laminopathies